# A linear tessellation model for the identification of "food desert": A case study of Shanghai, China

**Lu Wang**[1], **Yakun He**[2]*, **Zhonghai Yu**[1], **Hongrui Wang**[3], **Wenjuan Ye**[4], **Xin Li**[1], **Yingping Liu**[1], **Junxiao Zhang**[5]

1 Jinan Surveying and Mapping Research Institute, Jinan, China, 2 Chinese Aeronautical Establishment, Beijing, China, 3 Zhejiang Academy of Surveying and Mapping, Hangzhou, China, 4 Shanghai Surveying and Mapping Institute, Shanghai, China, 5 Faculty of Geosciences and Environmental Engineering, Southwest Jiaotong University, Chengdu, China

* hyk1990@whu.edu.cn

**Data Availability Statement:** The data underlying the results presented in the study are available from Github (https://github.com/wl7/A-linear-tessellation-model-for-the-identification-of-food-desert-A-case-study-of-Shanghai-China).

## Abstract

The "food desert" problem has been treated under a national strategy in the United States and other countries. At present, there is little research on the phenomenon of "food desert" in China. This study takes Shanghai as the research area and proposes a multiscale analysis method using a linear tessellation model that splits the street network into homogeneous linear units. Firstly, the network kernel density estimation using a linear tessellation model is used to measure the travel-mode-based food accessibility. Considering the actual travel constraints, the GPS trajectory data of four travel modes (walking, bicycle, metro and taxi) are applied to calculate the speed of each linear unit. Secondly, the "food desert" phenomenon in Shanghai are identified combing with the results of the network K-function. Finally, the resident income conditions in different modes are fitted based on the housing price data and the spatial distribution of four "food desert" patterns are detected by the overlay analysis of food accessibility and resident income conditions. The experimental results show that fifty percent of Shanghai is characterized by low food accessibility, and half of these areas are disadvantaged and low-income areas in suburbs, which are the locations experiencing the "food desert" phenomenon. Comparing the results of the proposed method and that of the traditional planar method, the identification results for all modes based on the traditional planar method underestimate the severity of the "food desert", especially for the bicycle and taxi modes. This study also provides corresponding decision-making reference for the alleviation and resolution of "food desert" issues. Moreover, the proposed method provides a new research perspective for urban research under the street network space.

## 1 Introduction

### 1.1 Background

According to the research of U.S. Department of Agriculture, limited access to healthy food and poor diet are the major pathogeneses of obesity, heart disease and related chronic diseases

**Funding:** This work was supported by the Shandong Provincial Natural Science Foundation [grant number ZR2024QD123]; the National Natural Science Foundation of China [grant number 42001336]. The funder of the Shandong Provincial Natural Science Foundation [grant number ZR2024QD123] is the first author of this paper, Lu Wang, who was mainly responsible for experimentalizing, writing original draft, revising and so on. The funder of the National Natural Science Foundation of China [grant number 42001336] is the 8th author of this paper, Junxiao Zhang, who was mainly responsible for revising the manuscripts and so on.

**Competing interests:** The authors have declared that no competing interests exist.

and thus have attracted the attention of the government and the public [1, 2]. More and more urban planners and geographic analysts have studied the relationships between food access, neighborhood socioeconomics and public health [3–5].

In the middle of 1990s, researchers defined "food desert" as the area where the inhabitants have limited access to healthy and affordable food within a certain distance due to the lack of adequate healthy food outlets [6]. From this perspective, the inherent characteristic of "food desert" is spatial. Many researches have started concentrating on phenomenon and defining the area of poor food access [7, 8]. With the advancement of data acquisition capabilities and computing capabilities, the actual conditions such as transportation systems and space-time constraints can be observed and quantified. This makes it necessary to develop more reliable and more fine-grained measures to measure healthy food accessibility.

The "food desert" problem has been treated under a national strategy in many countries worldwide. In the field of urban study, many researches have investigated the "food desert" problem and related social inequality in developed countries [9–12]. At present, there is little research on the phenomenon of "food desert" in China [13].

## 1.2 Literature review

Most previous studies have focused on exploring "food desert" by analyzing the following four regional characteristics: (1) access to food stores [13, 14]; (2) regional racial/ethnic differences [15, 16]; (3) regional income/social-economic status [17, 18]; (4) differences in store types and food supply [19, 20]. There is no agreement on how to identify the "food desert", but food accessibility analysis has always been the preferred method because of its quantitative characteristics [21]. The traditional methods for accessibility analysis of food accessibility consist of density-based methods and proximity-based methods [22].

The density-based methods quantify the accessibility of food outlets with total count, count per population or count per square area using the buffer method, kernel density estimation (KDE) or spatial clustering. As one of the most common methods for measuring the first-order effects of a spatial process in spatial point patterns, the KDE method is a recommended geographic method for establishing access to amenities in health research [23]. This method can transform the distribution of point events into continuous surfaces with a quartic kernel and an adaptive bandwidth, and examine the intensity trend of events in space, and it has been applied to studies of food access [24]. Numerous researchers utilize KDE to measure the density of various food outlets to assess the food environment around homes, work sites, schools and other destinations [3, 4, 24–26]. Then, the disparities in the food environment are analyzed based on the KDE results and selected neighborhood characteristics, such as poverty rates, unemployment rates, vehicle ownership rates, public transit access, and other socio-economic indicators [26]. Conforming to Tobler's First Law [27], the KDE method not only takes the number of features nearby into account, but also applies a weighting function so that the weight of the nearest facility is greater than that of more distant facilities [28].

As a well-known method of second-order effects in spatial point patterns used to examine the correlation and independency of point features, the K-function statistical method is also applied to measure food environment [23, 29]. Based on this method, Austin et al. [30] and Day et al. [31] examine the clustering of fast-food restaurants in areas proximal to schools to characterize schools' neighborhood food environments. Furthermore, Costa et al. [32] measure access to commercial fruit and vegetable establishments with KDE and the K-function. Most studies regard the block or neighborhood as the basic unit when applying these methods, which may lead to the edge effects by restricting human activities within established administrative units [24, 25, 33–35]. Furthermore, these results possibly neglect differences in

accessibility within the same administrative unit and increase distortion of the estimation [36]. To address this issue, geographic grids have been utilized to refine the analysis unit [37]. They have two advantages: (1) the geographic grid structure can describe data distribution more accurately; (2) the size of each grid unit can be adjusted to satisfy different analysis demands and analysis factors. Nevertheless, the abovementioned methods regard the region as homogeneous and isotropic Euclidean space, ignoring the fact that human activities are restricted to street network space [38]. This may result in an inaccurate estimation of accessibility between the study region and food outlets because of the false cluster patterns in planar space [39].

Aiming at solving the problems above, proximity has been proposed to estimate the "food desert" phenomenon by the distance or the travel time between food outlets and the study region with the network analysis model [40]. Based on this model, many scholars analyze the food environment of the study unit by measuring the shortest network distance between the centroid of the study unit and the nearest food outlets [41]. However, for numerous reasons, people do not always acquire food from the closest stores [42]. Burns and Inglis [43] utilized the same method to conduct a comprehensive analysis according to the traffic modes, road types and other characteristics of the traffic network (i.e., the frequency of buses). Nevertheless, the above studies treated each road as a homogeneous geometry arc with fixed length [44]. Abundant data enriches the factors considered in food accessibility analysis, such as real-time travel burden [45], transport costs [46], transport modes [13], and the accessibility of transportation [47]. Although these studies provided a better description of food accessibility, they still remained at administrative level analysis, failing to measure the accurate individual accessibility.

Combining the characteristics of the KDE method and the network analysis model, the network KDE method was developed by Borruso [48] to calculate the density based on the shortest network distances. It can identify the aggregation of network-constrained phenomena more accurately. Tang et al. [28] applied network KDE to analyze the space–time distribution of taxis' pick-up events. Referring to planar K function, Okabe and Yamada [49] proposed the network K-function. Yamada and Thrill [50] proved that planar K-function analysis is problematic because it entails a significant chance of over detecting clustered patterns by comparing the results based on the network and planar K-function methods. Enlightened by network-based methods, researchers have made efforts to analyze network-constrained point events under network space instead of planar space, such as retail stores [51], traffic accidents [36], crime [52] and other events associated with economic activities [38].

In addition to poor food accessibility, low income is a basic characteristic of "food desert" areas. Because, it has been widely utilized "food desert" to describe areas where low-income residents do not have access to healthy and affordable food [1, 7]. Based on this description, the areas are identified as food deserts if they meet both low-income and low-food-access criterion in many studies [53, 54].

Based on the above analysis, the geographic grid structure can be adjusted to satisfy different analysis demands and analysis factors. The network arc structure can make the accessibility analysis fully consider that human activities rely on connections in the traffic network and is limited by the space-time constraints of the street network [55]. In order to combine the advantages of the above two methods, this study proposes a linear tessellation model to measure fine-grained accessibility in street network space. In traditional Euclidean space, the raster data divides the planar space into continuous regular cells based on the field theory, and the analysis scale is controlled by adjusting the size of regular grids. Similarly, the proposed linear tessellation model splits the street network into homogeneous linear units and the analysis scale of the network is also adjusted by setting the units' length which makes solving issues at a fine-grained scale possible [56].

### 1.3 Study objective

Based on the proposed linear tessellation model, this study attempts to: (1) introducing the network Kernel Density Estimation analysis method into the linear tessellation model to measure transit-varying food accessibility in Shanghai with trajectory data; (2) applying the network K-function method to analyze the estimated food accessibility from the perspective of food outlets' distribution pattern; (3) identifying the "food desert" areas by the food accessibility and income and discussing the implications for urban planning. The proposed model aims to provide an innovative thought with a rational analysis space for food acquirement issue, and to provide reasonable implications for public policymakers by exploring the state of "food desert" at a finer size.

In the next section, we demonstrate materials and methods of "food desert" identification based on the linear tessellation model. Then, our experimental results and discussions are presented in Section 3. Finally, Section 5 concludes this paper and summary the innovations and limitations in our study.

## 2 Materials and methods

### 2.1 Study area and data

Shanghai's rapid urbanization and expansion have been affecting the communication, travel, work and lifestyles of millions of residents. It displays the positive and negative outcomes of urban growth, which makes it a typical place for diverse urban research [57].

To apply the proposed method to identify the food accessibility in the Shanghai region as shown in Fig 1(A), where the urban center and urban fringe area is within the red line and blue line, respectively, the remaining area is suburb.

The study data are as follows:

1. Healthy food retail outlets. We obtained 12,419 records of the food retail outlets via the Baidu API (http://lbsyun.baidu.com/index.php?title=webapi/guide/webservice-placeapi), including full-service supermarket chains, fresh markets, sea food stores, fruit and vegetable grocery stores [13], as shown in the green points in Fig 1(B);

2. Entire road networks. These data are downloaded from OpenStreetMap (https://www.openstreetmap.org/), as shown in the grey lines in Fig 1(B);

3. House price Point of Interests (POIs). These data are obtained from the houses put up for sale in the Lianjia website (https://lianjia.com/) in March, 2017. The total number of points is 91,619.

4. Trajectory data. Allowing for the travel mode of residents and low ownership rate for private cars, four types of trajectory data, walk, bicycle, metro and taxi, are selected to calculate the average speed of each road segment for the corresponding travel mode. The estimated speed of the walk, bicycle, taxi and metro mode is shown in Fig 2(A)–2(D), respectively.

GPS trajectory data of walk and bicycle are obtained from Yidong GPS APP (http://edooon.com/app/gps/). GPS trajectory data for taxi are obtained from SODA (Shanghai Open Data Apps, http://shanghai.sodachallenges.com/data.html). All the GPS trajectory data are in the same time period (March 1 to 31, 2017). The speed of metro mode is set based on the average speed of 14 metro lines in Shanghai, as shown in the illustration in the lower right corner of Fig 1(B). The average speed of 14 metro lines is calculated according to the total metro lines' length provided by the Shanghai Survey and Mapping Institute (https://www.shsmi.cn/) and

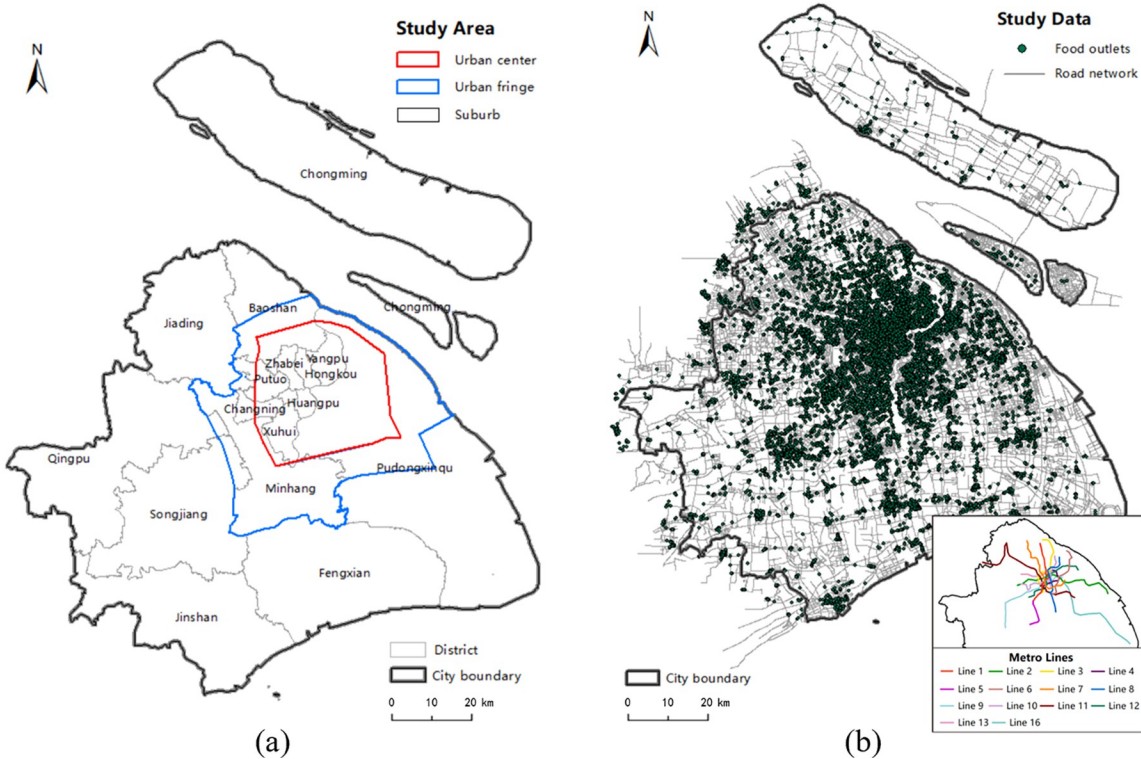

**Fig 1. The Shanghai region.** (a) presents the districts and planning of study area, where the black line refers to city boundary of Shanghai, the grey line refers to district boundary, the area inside the red line is urban center, the area outside the red line and inside the blue line is urban fringe, the remaining area is suburb. (b) presents the study data, where the green points refer to the food outlets, the grey lines refer to the road network, the illustration in the lower right corner shows metro lines. Date Source: Contains information from OpenStreetMap and OpenStreetMap Foundation, which is made available under the Open Database License.

the total metro operation time provided by the Shanghai Metro official website (http://www.shmetro.com/).

The house price POIs data and trajectory data are selected from the same time period, which ensures the effectiveness and reliability of our study. The location attribute of house price POIs contributes to obtain finer-scale characteristics of resident instead of the characteristics of whole administrative units. And the trajectory data has ability to show real-time traffic conditions of each road segments with calculated speed, further to simulate space-time constraints encountered by humans in the road networks.

## 2.2 Methods

The identification of "food desert" based on the linear tessellation model can be divided into five steps: (1) generation of the linear tessellation model based on four transport modes trajectory data and road network data; (2) measurement of food accessibility by the network KDE method based on the established linear tessellation model; (3) analysis of food environment from the perspective of food outlets' distribution pattern by the network K-function method based on the established linear tessellation model; (4) calculation of resident income from house price with the housing price income ratio indicator; (5) identification of "food desert" areas with both low accessibility and low income criterion. The general process is described in Fig 3.

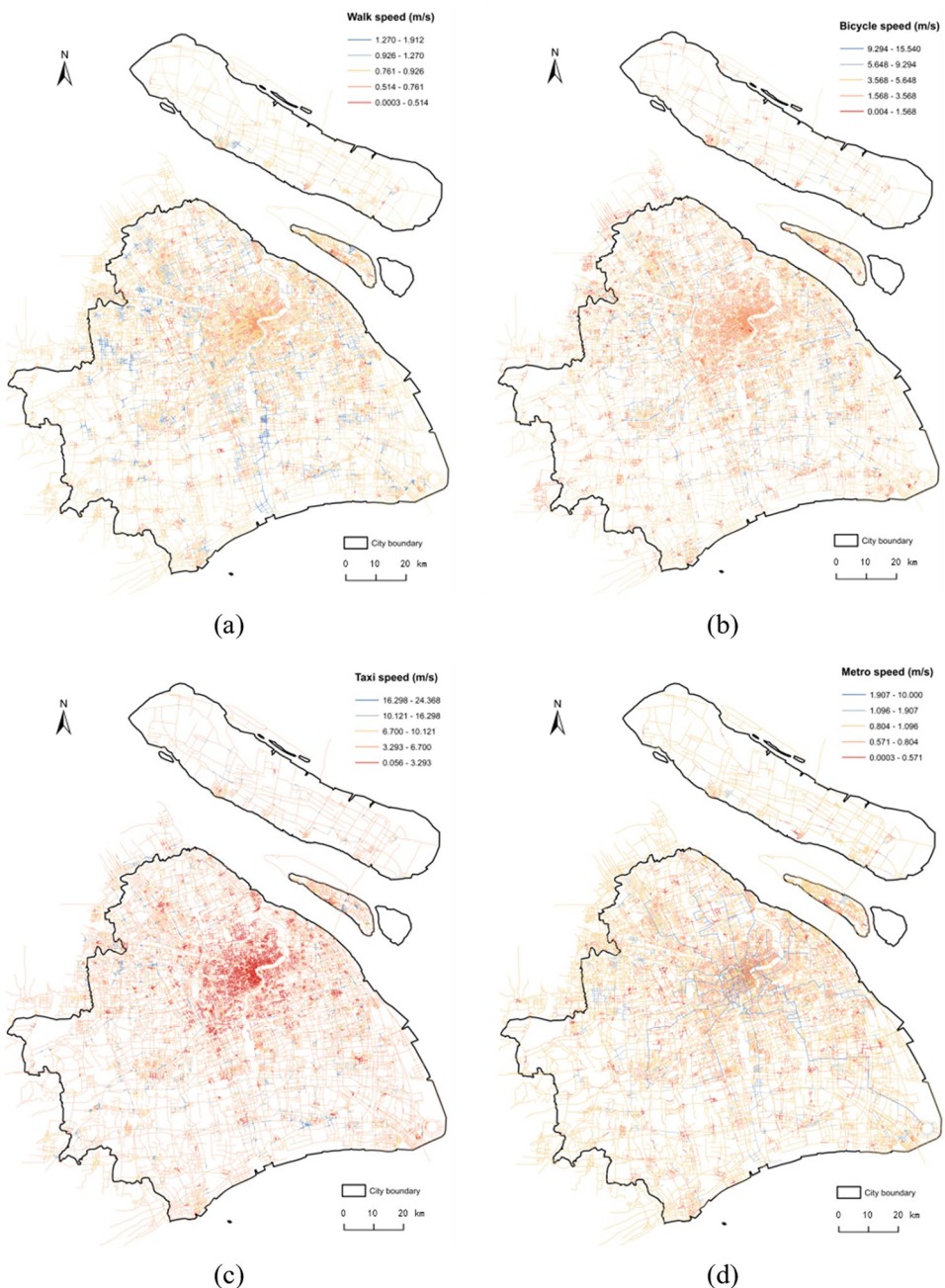

**Fig 2. The speed map of the four transport modes calculated from the trajectory data in road network.** (a) presents the average speed of each road segment for the walk mod, where the speed values range from 0.0003 to 1.912. (b) presents the average speed of each road segment for the bicycle mode, where the speed values range from 0.004 to 15.540. (c) presents the average speed of each road segment for the taxi mode, where the speed values range from 0.056 to 324.368. (d) presents the average speed of each road segment for the metro mode, where the speed values range from 0.0003 to 10.000. Date Source: Contains information from OpenStreetMap and OpenStreetMap Foundation, which is made available under the Open Database License.

**2.2.1 Linear tessellation model.** For traditional Euclidean space, the raster data divides the plane into continuous regular cells based on the field theory. The value of each cell and the neighborhood relationship among the cells are recorded in the planar tessellation model of

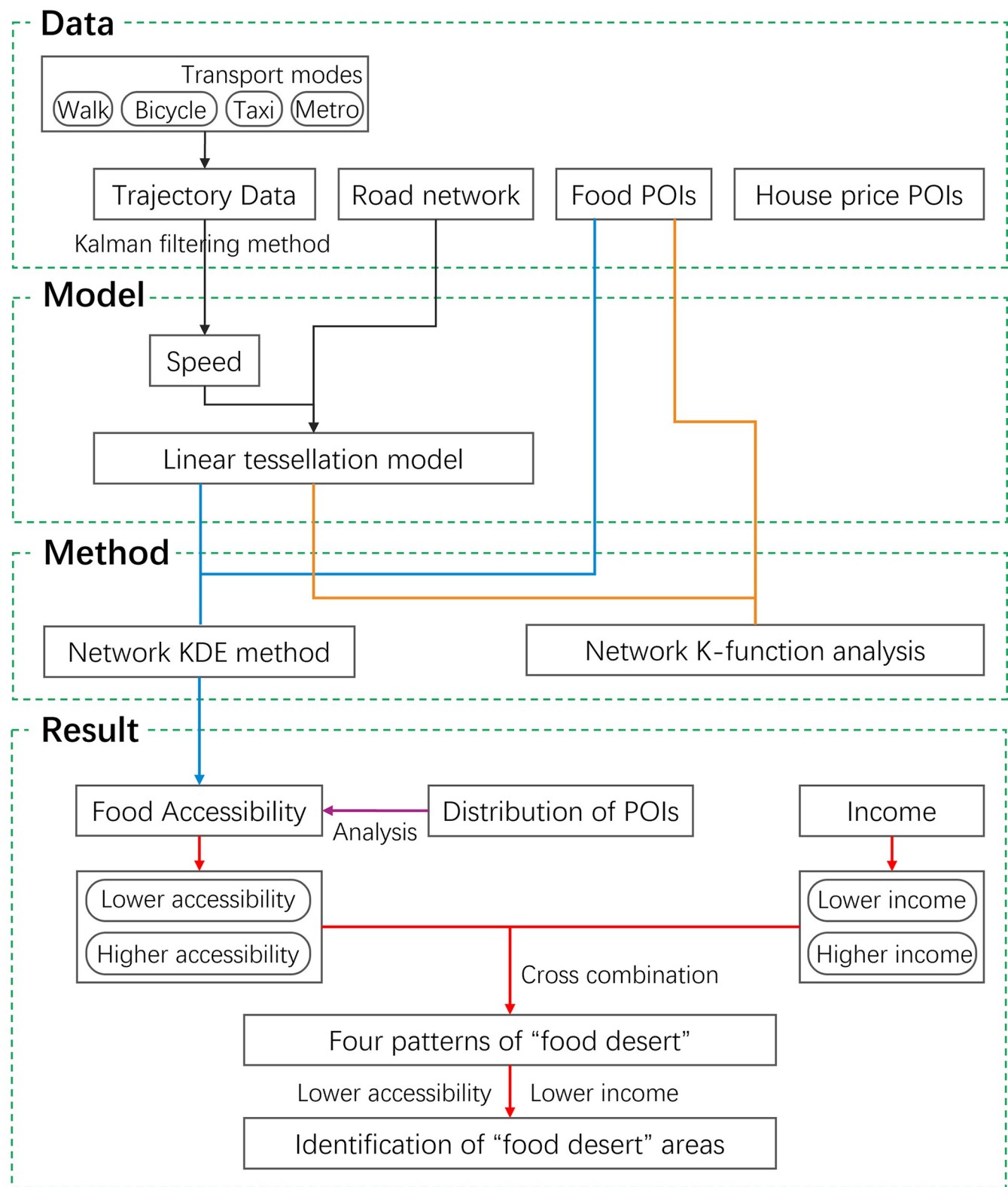

**Fig 3. Overall framework for identification of "food desert" areas.** The black flowlines refer to step (1), the blue flowlines refer to step (2), the orange flowlines refer to step (3), the purple flowline refers to step (4), and the red flowlines refer to step (5).

raster data, which enables efficient spatial analysis in the planar space. This method can also be applied to the network space. The linear tessellation model is proposed to split the street network into a set of homogeneous basic linear units with adjustable length according to the analysis factors. Then, a linear reference system is generated using basic linear units and the topological relationship (including the adjacency relation between linear units and the inclusion relation between the intersections of adjacent basic linear units and linear units), where the distribution law of events within network space can be explored using the basic linear unit as analysis unit.

In Euclidean space, the analysis scale is controlled by adjusting the size of regular grids. By analogy, the analysis scale of the network space is also adjusted by setting the length (namely, the basic linear unit length of an arc), which depends on the defined length and the weight of partition. The form of partition length is:

$$L_i = l_e \cdot f_i \tag{1}$$

where $L_i$ is the length of basic unit for $i^{th}$ road; $l_e$ is the defined length; $f_i$ is the weight function.

To better simulate human behavior in the network space, each road segment receives different speed-based weight values in term of the corresponding transport mode and real-time speed. Therefore, the weight function $f_i$ can be represented as:

$$f_i = \begin{cases} \dfrac{v_i}{l_e} & v_i > 0 \\[2mm] \dfrac{v_{\min}}{l_e} & v_i = 0 \end{cases} \tag{2}$$

where $v_i$ is the average speed (m/s) of the $i^{th}$ road; and $v_{\min}$ is the minimum speed (m/s) in human travel. The average speed of each road segment is calculated by trajectory data with Kalman filtering method [58]. Because Kalman filter can estimate the position and speed of a vehicle from a series of incomplete and noisy data, which makes it an optimal estimation method and is widely used in the field of automatic driving. Based on this speed-based weight, the road network is divided into a series of isochronous basic units with varying lengths, where each basic unit represents 1 second.

**2.2.2 Network kernel density estimation method based on the linear tessellation model.** The KDE is a spatial smoothing method employed to transform a sample of geographically referenced point data (e.g., address of food outlets) into a smooth continuous surface to estimate the "intensity of referenced points across a surface, by calculating the overall number of cases situated within a given search radius from a target point" [59]. Compared to other analysis methods, the KDE method can calculate the density at any position in the study region [23]. It can also visualize the phenomenon by means of a smooth and three-dimensional continuous surface, in which peaks represent the presence of clusters or 'hot spots' in the distribution of events (e.g., food outlet POIs), and any position will have higher density if it is at a shorter distance from the POIs. This method holds characteristics of Tobler's first law of geography [48], which makes it conform to the influence mode of events with distance decay. Applying all the advantages of the KDE method to calculate event accessibility, the general formula is expressed as:

$$A(s) = \hat{\lambda}(s) = \sum_{i=1}^{n} \frac{1}{\tau^2} k(x)$$

$$x = \frac{s - c_i}{\tau} \tag{3}$$

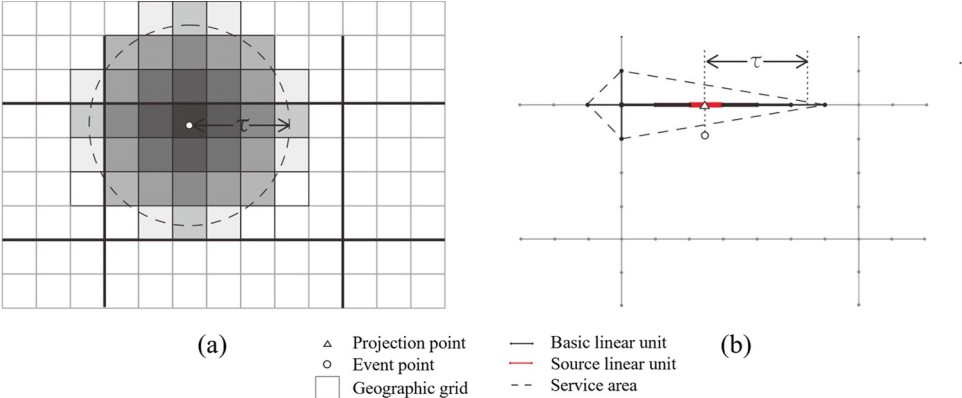

**Fig 4. Comparison between planar KDE method and network KDE method.** (a) presents the result of planar KDE method, where the darker the grid, the higher the KDE value. (b) presents the result of network KDE method, where the thicker the basic linear unit, the higher the KDE value. The areas within the dashed lines represent the service area of the event points.

where $A(s)$ is event accessibility value at position $s$; $\hat{\lambda}(s)$ is the estimated density value at position $s$; $k(x)$ is a weight function; $c_i$ is the $i^{th}$ event point; $s-c_i$ is the distance between event point $c_i$ and position $s$; $\tau$ is bandwidth; and $n$ is total number of event points under concern.

Assuming that the Euclidean space is continuous, homogeneous and isotropic, the traditional planar KDE method utilizes a weight function for any location with identical properties based on the Euclidean distance measurement [38]. It is easy to neglect spatial heterogeneity and network constraints, resulting in overestimate for network-constrained event accessibility [51]. Given abovementioned problems, the network KDE method applied to the linear tessellation model provides an urban environment analysis with a more fine-grained level, as shown in Fig 4.

The network KDE method is an application of the KDE method for network space. Therefore, the Eq 3 is applicable to the density calculation both of the network and planar KDE method. However, instead of the Euclidean distance measurement in planar method, the bandwidth and weight function are calculated by the shortest-path distance in network method. Considering that the proposed linear tessellation model has divided the road networks into a series of basic units with equal value (e.g., equal length and equal time), the bandwidth is transformed to the number of linear basic units as the threshold of search steps in the calculation. We utilize step-by-step expansion from the source linear unit projected by each event point to all directions of network space based on the topological relationship to search for the linear units within the threshold. And the shortest-path distance refers to the network distance between the centers of the two linear units along the corresponding expansion direction.

Therefore, the detailed procedure of network KDE method based on linear tessellation model is presented as follows:

1. Assign the event POIs (such as food outlet POIs) into the nearest basic linear unit in Euclidean distance, and define these basic linear units as the source basic linear units;

2. Expand the $i^{th}$ source basic linear unit by querying for the neighbouring basic linear units via adjacent node step by step, record the step frequency of each basic unit, expanding from source unit as the distance between them within step $\tau$.

3. Calculate the density of each basic unit according to Eq 3 within step $\tau$.

4. Expand all source basic linear units by repeating procedures (2) and (3) and use the sum of density calculated in each expanding process as accessibility for each basic unit.

Fig 5 shows the detailed process of search and calculation for one event point in the case of Fig 4(B), where the basic units are equal length of 10m, the number of search steps is 3. Firstly, the event point is assigned into the basic linear unit *o* as the source unit, as presented in Fig 5(A). Then, expand the 1st step from source linear unit *o* by querying for its neighboring linear units. It is easy to obtain the units *a* and *b*, which are marked in Fig 5(B). Record the step frequency as 1 and the shortest-path distance to unit *o* as 10m for units *a* and *b*. In the second step of expansion, the linear units *c* and *d* (Fig 5(C)) can be searched by querying for the neighbors of units *a* and *b* with 1 step frequency, respectively. Record the step frequency as 2 and the shortest-path distance to unit *o* as 20m for units *c* and *d*. With the same operation, as displayed in Fig 5(D), the units *e*, *f*, *g* and *h* can be obtained in the last step of expansion, where their step frequency is recorded as 3, and the shortest-path distance to unit *o* is 30m. According to Eq 3, calculate the density of units *a*, *b*, *c*, *d*, *e*, *f*, *g* and *h*.

In the network KDE analysis based on the linear tessellation model, the description for accessibility does not depend on traditional shortest path analysis, and it consists of all possibilities to acquiring food related to the density of both the food outlets and the road network.

It is generally agreed that the selection of the bandwidth value is vital for KDE analysis in terms of determining the appropriate spatial scale for different circumstances, and this value affects the statistical results by determining the smoothness of the KDE value [38, 48, 51]. Similarly, for parameter setting in network KDE method based on proposed linear tessellation model, the threshold of search steps is more important than weight function $k(x)$. Therefore, a quartic function is adopted as the weight function in our experiments of the planar method and the network method, which is often utilized for density calculation. The formula is as

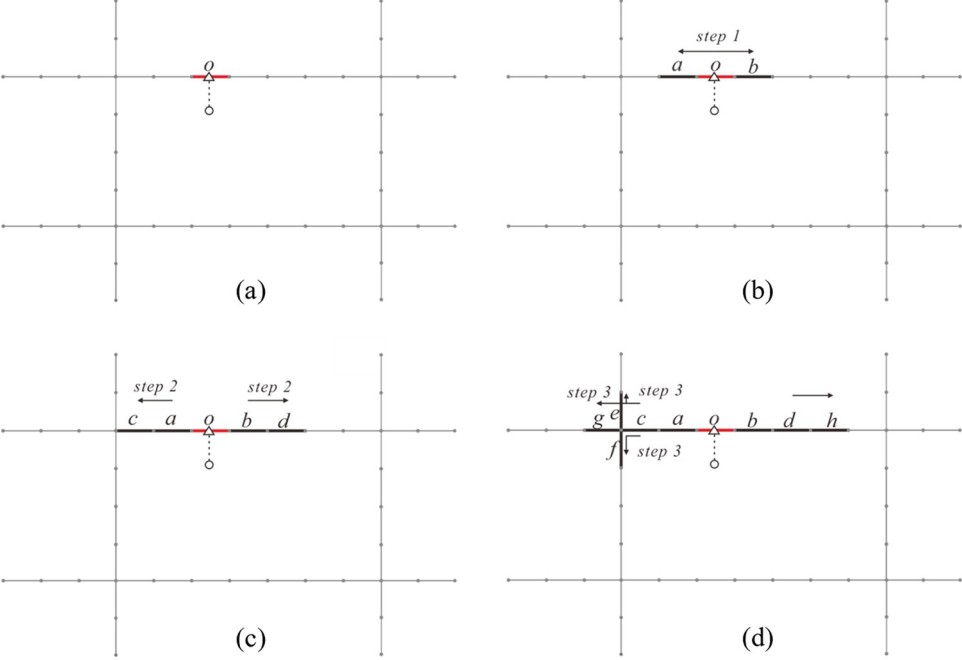

**Fig 5. The calculation procedure of the network KDE method based on linear tessellation model.** (a) presents the determination of the source basic linear unit. (b) presents the first step of expansion. (c) presents the second step of expansion. (d) presents the third step of expansion.

**Table 1. Parameters of KDE analysis in planar and network methods.**

| travel mode | average speed (m/s) | bandwidth | |
|---|---|---|---|
| | | planar (m) | network (s) |
| walk | 0.8 | 1440 | 1800 |
| bicycle | 3.6 | 3240 | 900 |
| taxi | 4.3 | 2580 | 600 |
| metro | 1.1 | 1320 | 1200 |

follows:

$$k(x) = \frac{3}{4}(1 - x^2)$$
$$x = \frac{s - c_i}{\tau}$$

(4)

Table 1 shows the setting of bandwidth of planar and network KDE method in four transport modes. Applying the KDE method to identify the "food desert", we define the area vulnerable to the phenomenon of low food accessibility: >30 min (walking), >15 min (bicycle), >10 min (taxi), and >20 min (metro) on the strength of the definition of "food desert" and the international standard for public facilities accessibility (Community Guide http://archived. naccho.org/topics/HPDP/commguide/) [13]. Based on the generated linear tessellation with isochronous units in section 2.2.1, the threshold of search steps in network method is represented by the limited time standard, which are 1800, 900, 600 and 1200 for walk, bicycle, taxi and metro mode, respectively. Then the length of bandwidth in network method is the maximum of all shortest-path distance from the source linear units to the units within threshold, which varies with the different event points and transport modes. Therefore, we utilize the time standard to express the bandwidth for network method. In line with the bandwidth of the network KDE analysis, the selection of bandwidth in the planar KDE analysis depends on the corresponding average speed and limited time of the four travel modes in the network. Additionally, to ensure the experiment efficiency and sufficient granularity of the results, we selected 100m as the defined length and the average speed of the walk mode, 0.8m/s, as $v_{\min}$ for the linear tessellation model after several rounds of experiments.

**2.2.3 Network K-function analysis method.** The K-function analysis method is regarded as one of the most effective and most widely applied methods in point pattern analysis because it examines point patterns at various spatial scales in consideration of all event–event distances rather than the distance from the nearest point [29]. This method can accurately identify the geographical level at which the points are most significantly clustered or dispersed. Making allowances for the spatial point events related to human activities that are constrained by a road network in real life, Okabe and Yamada [49] propose the network K-function method, an extension of the planar K-function.

As defined, for a set of points $P = \{p_1, p_2, L, p_n\}$ distributed on street network $L_T$ with length of $l_T$, the expected network K-function under complete spatial randomness (CSR) [60] can be defined as:

$$K_{Exp}(h) = \frac{1}{\rho}E(h, P, L_T)$$

(5)

where $h$ is the density of points in $L_T$, calculated by $\rho = n/l_T$; $E(h, P, L_T)$ is the expected number of points within network distance $h$ to a point in $P$ on $L_T$. The expected number of points within $h$ to a point in $P$ over network $L_T$ is estimated under a Poisson process.

The observed network K-function of place $p_i$ represented as following formula:

$$K_{Obs}(h) = \frac{1}{\rho} \frac{\sum_{i=1}^{n} N(h, p_i, L_{Ti(h)})}{n} \tag{6}$$

where $L_{Ti(h)}$ is the subset of the network that consists of point within a network distance $h$ of $p_i$ in $P$; $N(h,p_i,L_{Ti(h)})$ is the total number of points in the subset $L_{Ti(h)}$.

To describe the distribution of the $n$ points in the network, Monte Carlo simulations are performed to generate a series of random distributions of $n$ points to obtain $K_{Exp}(h)$ value and the upper and lower envelopes of CRS for a given network. If $K_{Obs}(h)$ is in the range of the CSR, the point set $P$ is in a random distribution, where each point has an equal opportunity to occur at any location in the network; the presence of one point at a certain location does not impact the probability of other points occurring at any location in the network. If $K_{Obs}(h)$ is above the upper envelope of CSR, the point set $P$ is in a clustered distribution. If $K_{Obs}(h)$ is below the lower envelope of CSR, the point set $P$ is in a dispersed distribution.

For the estimation of the planar and network K-function and their statistical significance, two widely used software packages were utilized: the Spatstat of R language for the planar K-function [61] and the SANET [62] tool for the network K-function. Given the excessive calculation (especially for the network K-function) and distribution characteristics of mass store POIs [39], all store food outlets are aggregated into 805 points based on the method of point cluster simplification with spatial distribution properties preserved [63].

## 3 Result and discussion

### 3.1 Comparison of estimated KDE value between planar and network method

Fig 1 shows that the majority of food outlets are concentrated in the central city and in the urban fringe, with a very few located around major roads and intersections in urban fringe. The estimated KDE values under the planar and network space for the walk, bicycle, metro and taxi transport modes are displayed in Figs 6–9, where the property of "none" presents low food accessibility with a value of zero.

Being equal to the proportion of area with none-zero KDE value, the scope of accessibility is an expression for service scope of food outlets within a defined time or distance threshold via different modes. Table 2 shows the calculated results of all types of transportation. From the perspective of value range, the planar method is likely to overestimate the concentration of food activities in comparison to the network method under the walk, metro and bike modes, but it is almost equal to the network method under taxi mode. From the perspective of scope of the accessibility, the region of low accessibility calculated by planar method is far larger than that identified by the network method.

In fact, the analysis results of the planar KDE method for the walk, bicycle and taxi transport modes simply reflect the performance of different bandwidth values. It is clear that from metro (Fig 9(A)), walk (Fig 6(A)), taxi (Fig 8(A)) to bicycle (Fig 7(A)) modes, the bandwidth becomes larger, the density surface becomes smoother and the scope of KDE result also becomes larger (Table 2). The distribution of the KDE values under the four transport modes all shows a core-periphery pattern with higher values in the more central neighborhoods, where the higher value clusters are mostly located in southeast of Putuo District, south of Zhabei District, central region of Hongkou, Huangpu and Yangpu District, with the remaining clusters being scattered around the urban fringe. It suggests that food accessibility estimated

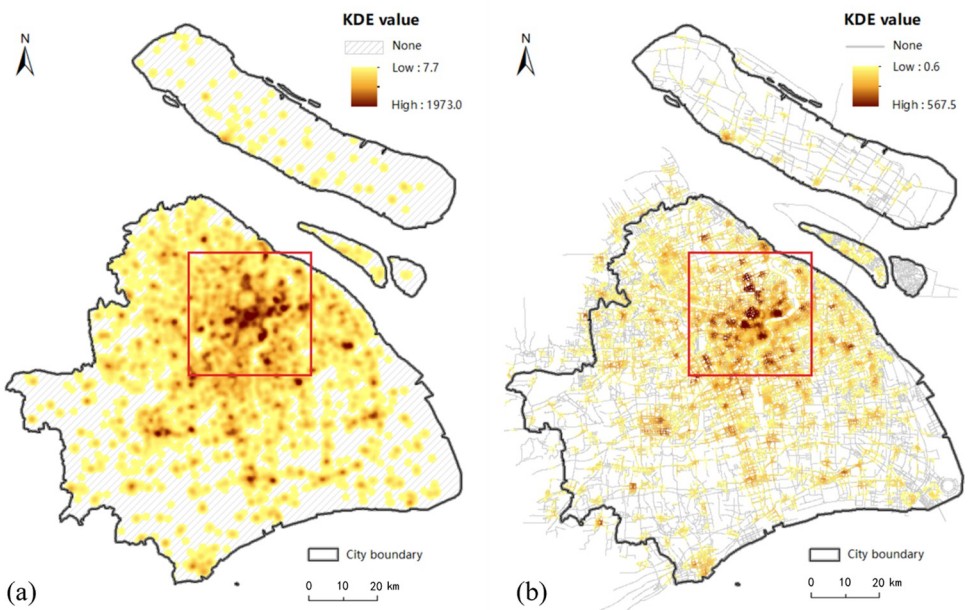

**Fig 6. The estimated KDE value of walk mode.** (a) presents the estimated planar KDE value of walk mode, where the KDE values range from 7.7 to 1973.0. (b) presents the estimated network KDE value of walk mode, where the KDE values range from 0.6 to 567.5. Date Source: Contains information from OpenStreetMap and OpenStreetMap Foundation, which is made available under the Open Database License.

by the planar method in all transport modes just depends on the distance from food outlets. And the planar KDE method can only simulate the "distance-decay effect" of the service capacity of food POIs.

As shown in Figs 6(B), 7(B), 8(B) and 9(B), as the bandwidth increases from taxi, bicycle, metro to walk mode, the density value becomes smoother and the scope of KDE value becomes larger based on the network method, which is consistent with the conclusion of planar method. However, different from the similar distribution pattern in the planar method, the estimated KDE values of four transport modes based on network KDE method have different distribution characteristics. Only in the walk and metro modes do the network KDE analysis results reveal a smooth core-periphery pattern similar to that of planar analysis. In particular, the network KDE result of walk mode has the distribution pattern most similar to the corresponding result for the planar space by visually comparing Figs 6 and 9. In the central areas with red outlines displayed in Fig 6, the higher KDE values of walk mode in both planar and network space are clustered in a large area. As for the metro-based results, the higher value clusters in the network result are more scattered in central city compared with planar result, as shown in the red outlines of Fig 9. From the network results of bicycle and taxi modes, it is clear that higher values are no longer concentrated in the central city, but they are dispersed along the network (Figs 7(B) and 8(B)), which is completely different from the planar results. Similar to the distribution of KDE values in planar results, for all transport-based accessibility results, low access values identify places where food outlets do not exist and road infrastructure is limited, especially in the fringe of the south and the north.

Compared the similar distribution of KDE value among the four modes based on the planar method, the differences in the network-based results proves that food accessibility is influenced by the space-time constraints calculated by the trajectory data, not just the distance from food POIs. Furthermore, the differences between the results of the network KDE method

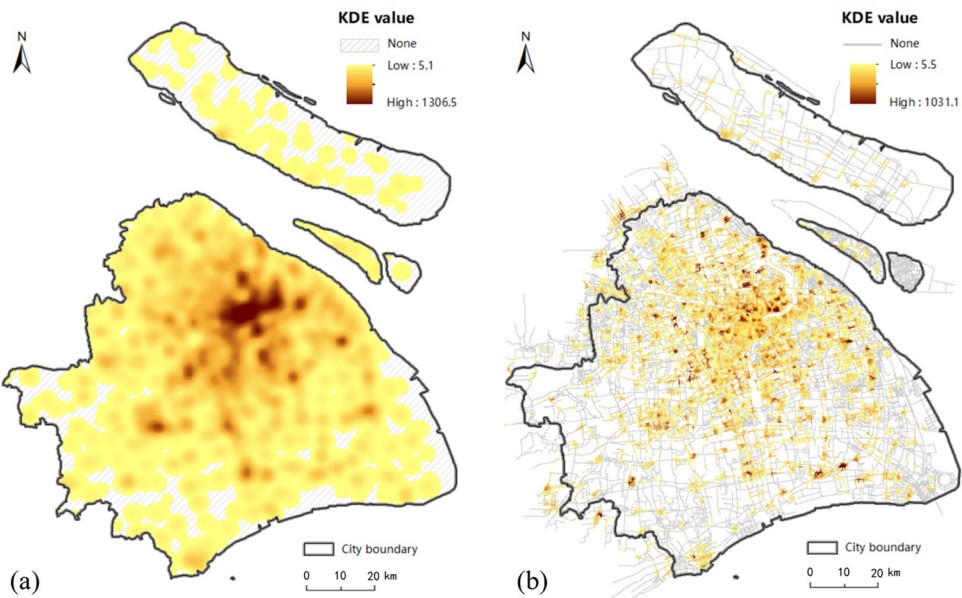

**Fig 7. The estimated KDE value of bicycle mode.** (a) presents the estimated planar KDE value of bicycle mode, where the KDE values range from 5.1 to 1306.5. (b) presents the estimated network KDE value of bicycle mode, where the KDE values range from 5.5 to 1031.1. Date Source: Contains information from OpenStreetMap and OpenStreetMap Foundation, which is made available under the Open Database License.

and the planar KDE method indicate two circumstances: 1) Bicycle-based and taxi-based food accessibility based on the network method are susceptible to two main external constraints: the real-time traffic condition and road planning. In the linear tessellation model, the speed of

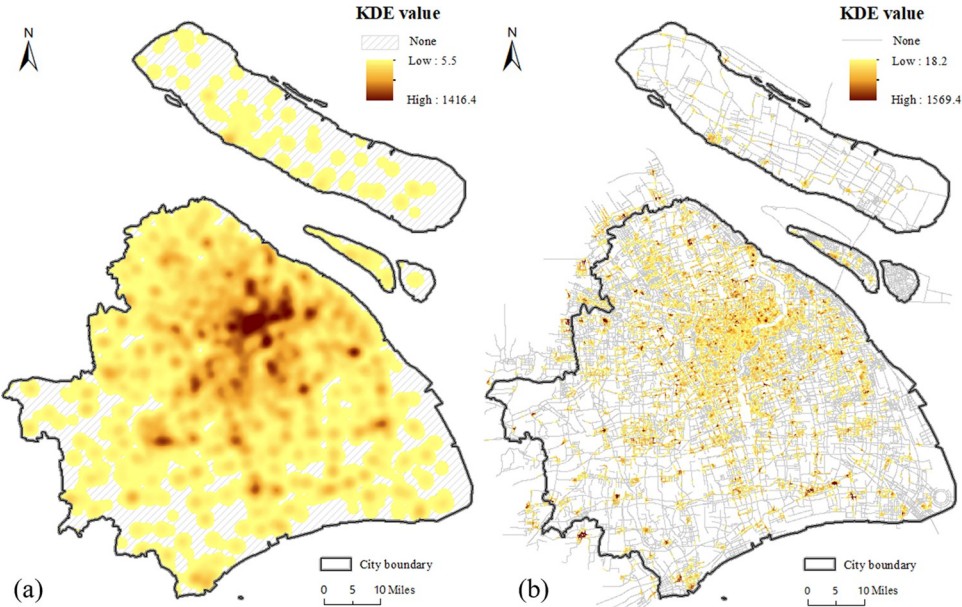

**Fig 8. The estimated KDE value of taxi mode.** (a) presents the estimated planar KDE value of taxi mode, where the KDE values range from 5.5 to 1416.4. (b) presents the estimated network KDE value of taxi mode, where the KDE values range from 18.2 to 1569.4. Date Source: Contains information from OpenStreetMap and OpenStreetMap Foundation, which is made available under the Open Database License.

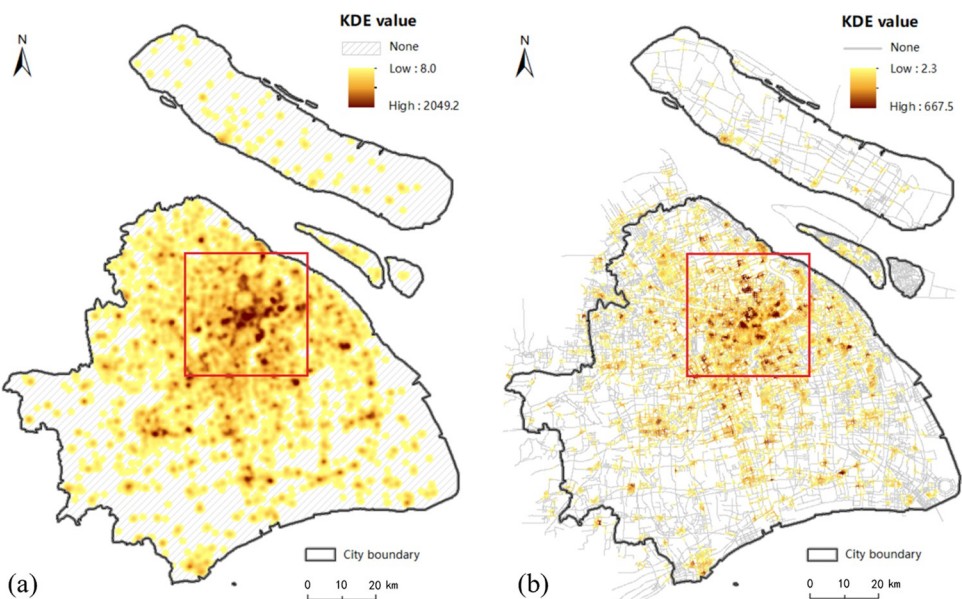

**Fig 9. The estimated KDE value of metro mode.** (a) presents the estimated planar KDE value of metro mode, where the KDE values range from 8.0 to 2049.2. (b) presents the estimated network KDE value of metro mode, where the KDE values range from 2.3 to 667.5. Date Source: Contains information from OpenStreetMap and OpenStreetMap Foundation, which is made available under the Open Database License.

each road segment is calculated by the trajectory data, which reflects the real-time traffic condition of the road segments. Besides, in these two travel modes, some road segments have no corresponding trajectory data. It is largely due to the impassability of bikes or taxis on these road segments, which is related to the road planning. 2) Although metro-based and walk-based estimated results are less affected by the real-time traffic condition, the difference of network results between these two modes described above indicates that accessing food by walking is mainly influenced by the distance from nearby retails, however, accessing food by underground depends on not only the location of food outlets but also other realistic factors such as the accessibility of metro stations and metro lines to the destinations for residents. Therefore, the network method can not only simulate the "distance-decay effect" of the service capacity of food outlets, which is also reflected by the planar KDE method. But it also simulates the real situations of each transport mode in the process of obtaining food within the road network, including the real-time traffic conditions, road planning and the accessibility of transport facilities. In conclusion, the network KDE method has a better ability to analyze food access which is a kind of human activities occurring in the road network.

**Table 2. The results of KDE analysis in planar and network methods.**

| travel mode | scope (%) | | value range | |
|---|---|---|---|---|
| | planar | network | planar | network |
| walk | 62.2 | 56.0 | 7.7–1973.0 | 0.6–567.5 |
| bicycle | 90.6 | 50.5 | 5.1–1306.5 | 5.5–1031.1 |
| taxi | 84.0 | 47.3 | 5.5–1416.4 | 18.2–1569.4 |
| metro | 58.9 | 51.8 | 8.0–2049.2 | 2.3–667.5 |

### 3.2 Analysis for the results of the planar and network K-function methods

As presented in Figs 10 and 11, the results of the K-function are represented by the cumulative number of points at a given distance, where the parameter *Obs* is the observed K-function, and the parameter *Exp(Mean)* is the expectation of random distribution. Furthermore, the parameters *Exp(upper 5.0%)* and *Exp(lower 5.0%)* are, respectively, the upper and lower envelope curves of the Monte Carlo simulation at a significance level of 0.05. The spatial pattern under analysis is characterized as "clustered" if observed curve is above expectation curve; "random" if observed curve nearly coincides with the expectation curve; "dispersed" if the observed curve is below the expectation curve; "significantly clustered" or "significantly dispersed" if the observed curve is outside of the confidence interval envelope of the CSR hypothesis.

To compare the spatial distribution patterns of food outlets, we analyze the planar and network K-function results based on the above spatial pattern characterization and the aggregation intensity for four areas: urban centers, urban fringes, suburbs and whole city (Fig 1(A)). Figs 10(A), 10(C) and 11(A), 11(C) show that the observed value (blue line) of planar K-function of all four areas is out of the upper envelope curve (green line) of the Monte Carlo simulation. It indicates that the food outlets in the planar space of four areas are all significantly clustered. However, the distribution patterns in the corresponding network space show obvious differences from those in planar space, especially in urban areas. As shown in the dashed lines of Fig 10(B), within 4.5 km, the observed curve of network K-function in urban center is

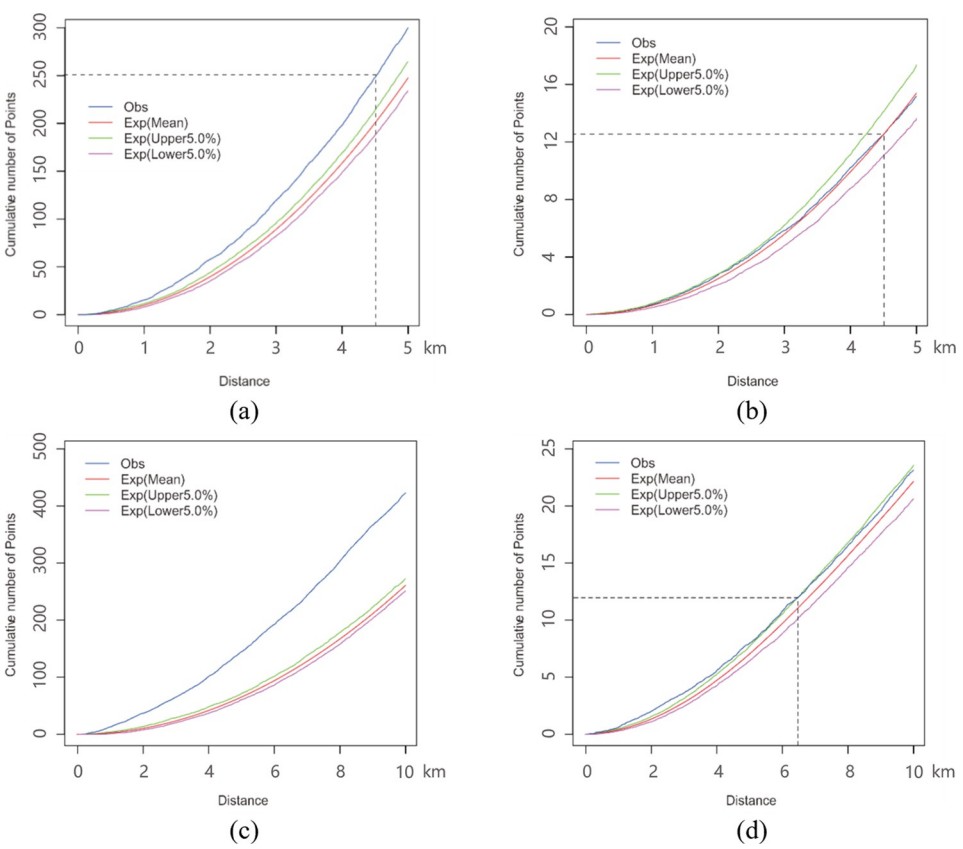

**Fig 10. The results of the K-function method within the urban center region and the urban fringe region.** (a) presents the planar K function curves of urban cente. (b) presents the network K function curves of urban center. (c) presents the planar K function curves of urban fringe. (d) presents the network K function curves of urban fringe.

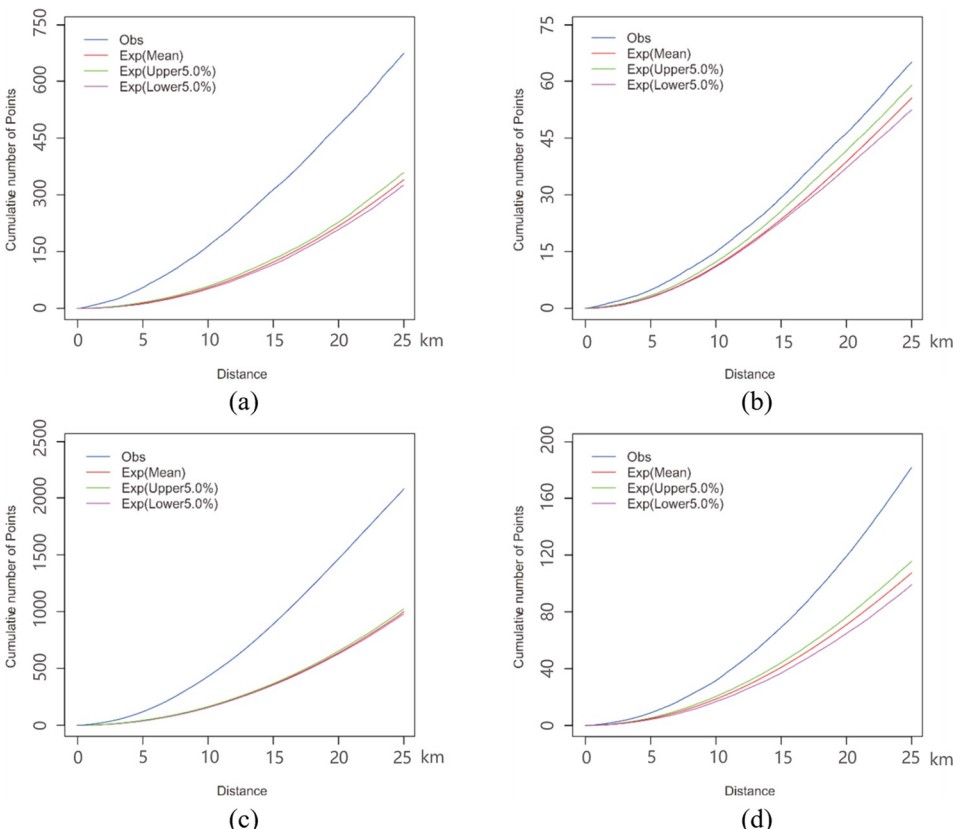

**Fig 11. The results of the K-function method within the suburb region and the entire city region.** (a) presents the planar K function curves of suburb. (b) presents the network K function curves of suburb. (c) presents the planar K function curves of whole region. (d) presents the network K function curves of whole region.

above expectation curve (red line) but below upper envelope curve, which implies that the food outlets in the road network of the urban center present an insignificantly clustered pattern. In 4.5–5.0 km, the observed value is below expectation curve but above lower envelope curve. Therefore, the food outlets are insignificantly dispersed in 4.5–5.0 km. However, on the whole, the observed curve nearly coincides with the expectation curve, which implies that the distribution of food outlets in network space is even in urban center area. Fig 10(D) shows that the observed curve nearly coincides with the upper envelope curve under the CSR hypothesis. Reflected by the dashed lines of Fig 10(D), the food outlets are slightly significantly clustered within 6.4 km, while they have a tendency towards insignificant aggregation over 6.4km in the urban fringe network. As seen in Fig 11(B) and 11(D), the K-function results in the suburbs and the whole city are that with the observed curve is out of upper envelope curve, inferring that the food outlets are significantly clustered in these two study areas.

Comparing the K-function results for planar and network space in each study area, the cumulative number of points in planar space is more than ten times higher than it in network space at the same distance. For example, in the 4.5 km of urban center region, the observed cumulative number of points is around 250 in planar result and 13 in network result, as shown in dashed lines of Fig 10(A) and 10(B). This difference of results between planar and network space also exists in the estimated number of fringe, suburb and whole city, which indicates the intensity of food outlet aggregation in planar space is far stronger than it is in network space. It is clear that the intensity of food outlet aggregation in both planar and network space become

stronger from center to fringe, with an increase in the cumulative number of points that reveals the distribution imbalance of food outlets between urban and suburb areas, particularly in network space. The random spatial distribution pattern in the urban center network space demonstrates that food outlets are evenly distributed in the road network, where food accessibility is relatively balanced for residents. Nevertheless, for the fringe region, the clustered pattern of food POIs results in food-access inequality to some extent. On the one hand, aggregated food outlets narrow the whole service scope, contributing to inaccessibility for people due to distance. On the other hand, the clustering of food POIs may result in crowded access behavior and further result in longer time to obtain food, contributing to inaccessibility for people in the aspect of time, especially for residents by car.

## 3.3 Identification and assessment for "food desert" phenomenon

According to the identification criterion, people in the "food desert" areas are of both low food accessibility and low income. Annual salary below 27,600 is defined as low income by the Shanghai Municipal Human Resources and Social Security Bureau [64]. Applying the finer income data instead of administration-based statistic data to the analysis, we utilize the house price POIs data to fit resident income with the housing price income ratio indicator [65]. The formula for income calculated by this ratio is expressed as:

$$I = \frac{P_h * A_h}{R} \tag{7}$$

where $I$ is per capita disposable income; $P_h$ is the average house price per square meter; $A_h$ is housing per capita; $R$ is the value of the housing price income ratio indicator. According to the "Statistical bulletin of Shanghai national economic and social development in 2017" from the Shanghai Statistics Bureau and the Yiju Real Estate College, $A_h$ is defined as 36.7 square meters, and $R$ is defined as 17.1. By employing the Kriging interpolation method, the calculated income result is mapped in Fig 12. It is no surprising that the higher income areas are in the more central neighborhoods, while the low-income areas mostly locate in the city fringe.

To further find areas where residents might be at risk of experiencing a "food desert" phenomenon, the estimated income is combined with the KDE value by spatial overlay analysis. Firstly, the Kriging interpolation method is utilized to calculate the KDE value of every location within the entire study area based on estimated network KDE value of four modes, as shown in Fig 13. This method can obtain the KDE value data suitable for spatial overlay analysis with income data. Although the Kriging interpolation results provides values for food accessibility in locations where addresses may not presently exist, it allows for an understanding of accessibility in areas of future development and provides a unique tool for visualizing accessibility across the whole city [10]. Then, we reclassify the interpolated KDE value data and income data to 0 and 1, respectively, where 0 represents low accessibility and low income, while 1 represents the high accessibility and high income, as shown in Table 3. Finally, we can obtain four patterns with cross combination, as shown in Table 4. The four patterns are named as HH, HL, LH and LL, corresponding to the risk level 0, 1, 2 and 3, respectively. The higher risk level represents the higher risk of "food desert". Residents in areas of pattern HH have higher access to food outlets and higher incomes, so they are not at risk of "food desert" phenomenon. On the contrary, the people who live in areas of pattern LL are characterized by low food accessibility and low income, which are exactly the characteristics of the "food desert" phenomenon. Comparing the explanation of pattern HL and pattern LH, people in areas of pattern HL have more access to healthy food, although they are lower-income. The areas of pattern LH are mainly located at the suburb fringe. As displayed in Fig 12, although they are

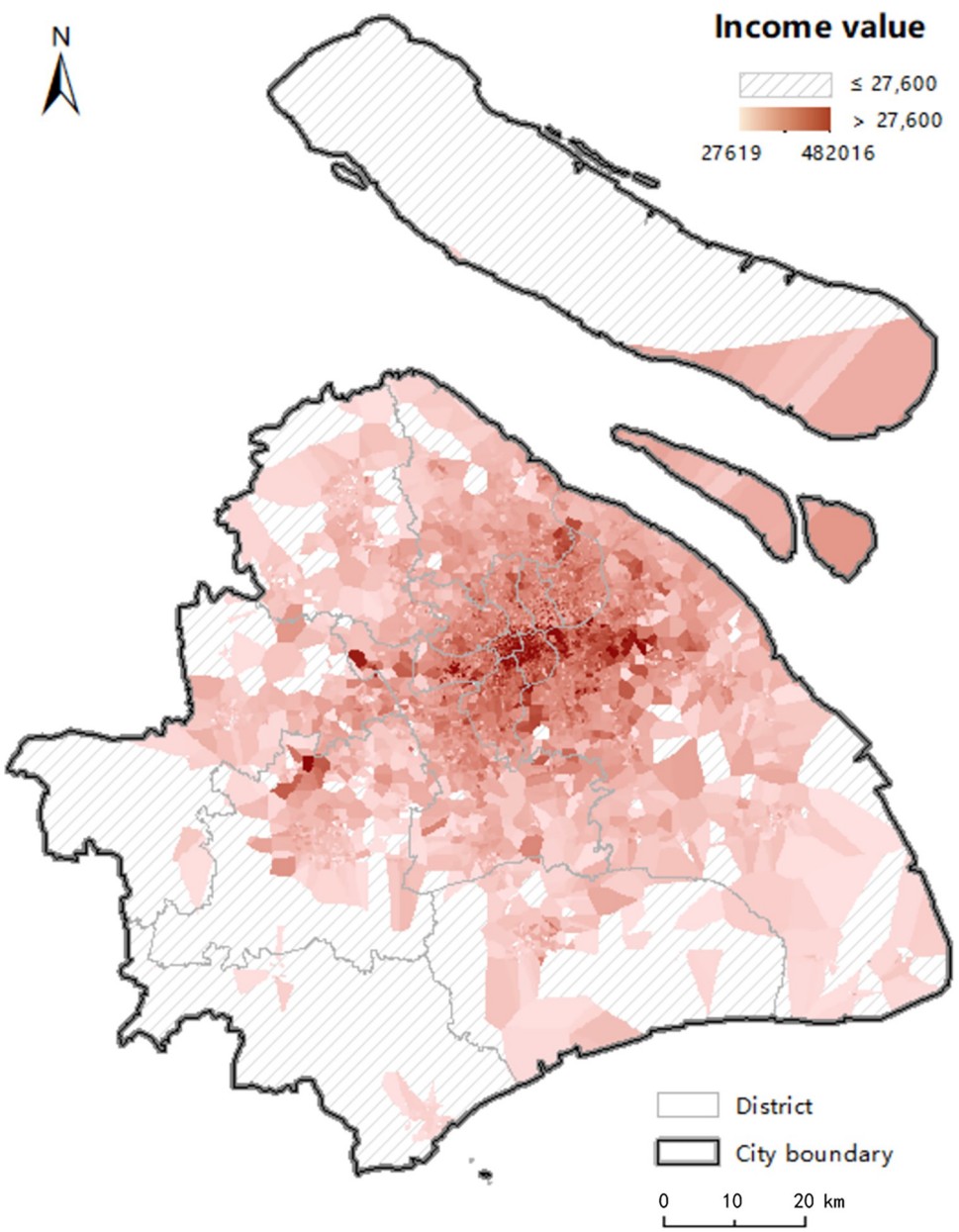

**Fig 12. The distribution of income calculated by the house price-to-income ratio.** The shaded areas indicate the income of residents is below 27,600. The colored areas indicate the income of residents is above 27,600, with the color getting darker referring the higher income. The shaded areas indicate the income of residents is below 27,600. The colored areas indicate the income of residents is above 27,600, with the color getting darker referring the higher income.

defined as "higher-income" areas, the incomes of residents in most of these areas slightly exceed the minimum wage standard. Therefore, people in pattern LH with low food accessibility are more vulnerable to "food desert" phenomenon compared with the people in pattern HL. Based on the analysis for the four patterns, we define pattern HH as the risk-free areas, pattern HL as the low risk level areas, pattern LH as the high risk level areas, and pattern LL as the top risk level areas which are the "food desert" areas.

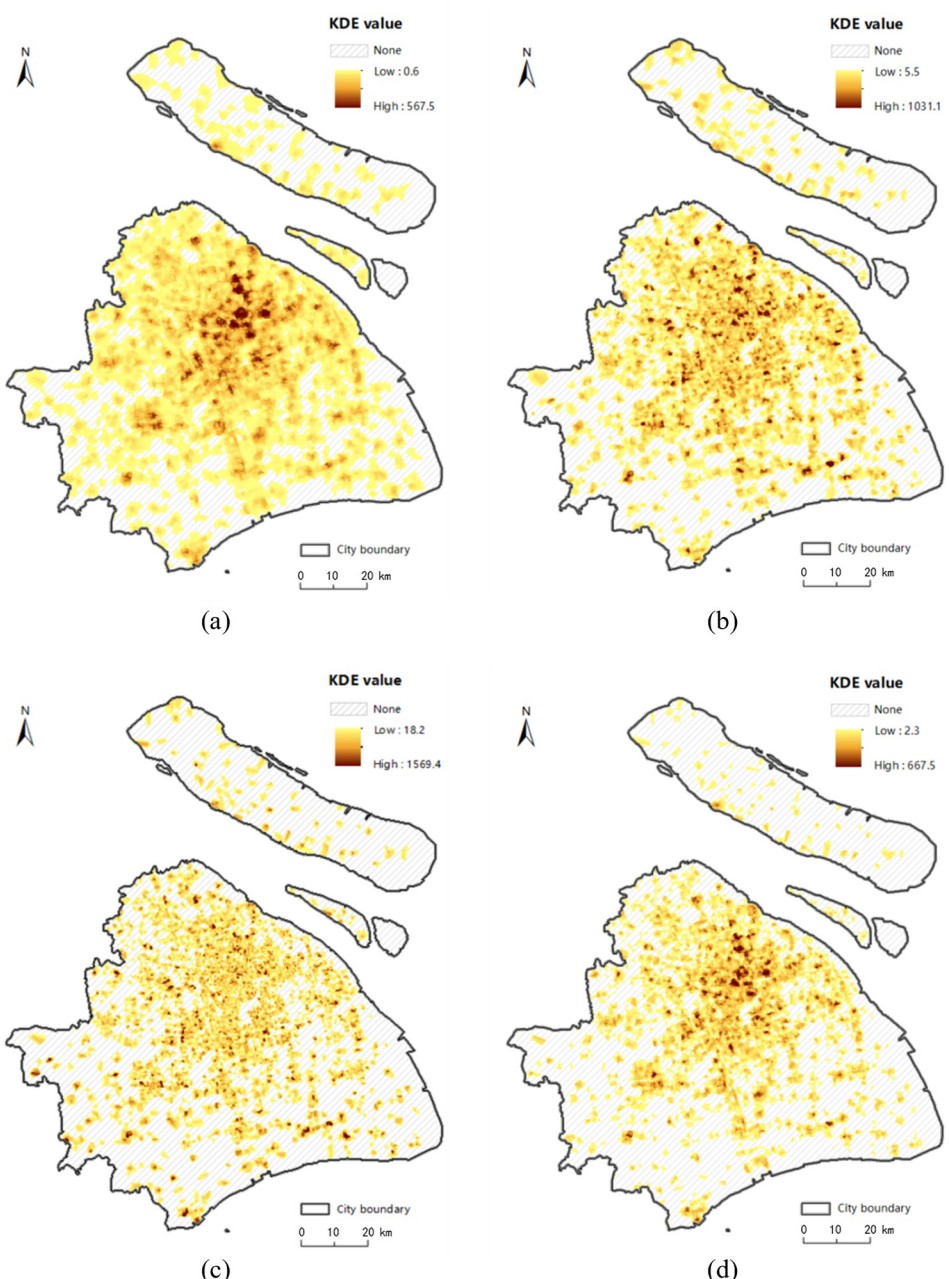

**Fig 13. The results of the Kriging interpolation method based on the network KDE value in four transport modes.**
(a), (b), (c) and (d) present the KDE values estimated by the Kriging interpolation method of walk mode, bicycle mode, taxi mode and metro mode, respectively, where the shaded areas indicate that the KDE values are equal to 0, the areas with darker color indicate that those KDE values are higher. Date Source: Contains information from OpenStreetMap and OpenStreetMap Foundation, which is made available under the Open Database License.

Table 4 displays the proportion of four patterns' areas for each transport modes based on planar method and network method. By analyzing the four patterns in general, there are great disparities between the estimated patterns with planar method and network method under bicycle and taxi modes. From the planar results of these two modes, the total proportions of

**Table 3. Reclassification of the KDE value and income value.**

| new value | original value | | explanation | |
|---|---|---|---|---|
| | **KDE value** | **income value** | **KDE value** | **income value** |
| 0 | = 0 | ≤27600 | Low food accessibility | Low income |
| 1 | >0 | >27600 | Higher food accessibility | Higher income |

patterns LH and LL of planar space are both approximately 10% (Table 4), which is far smaller than the same proportions in the network estimation. This shows that the identified risk level is much lower when the planar KDE method is applied; that is to say, the quantity and distribution of food outlets satisfy the majority of residents' demands for food access considering the aspect of distance but without taking into account the fact that the real process of acquiring food occurs in the road network. Additionally, the great disparities between the results of the planar method and network method under bicycle and taxi modes are mostly reflected in urban fringe and suburb areas (Fig 14). According to the network-based results in bicycle (Fig 14(B)) and taxi modes (Fig 14(D)), residents living in fringe areas are vulnerable to food inaccessibility when they choose these two modes. It is understandable that the insufficient but clustered food retails among the roads of fringe areas provide limited service capability and scopes for residents. Furthermore, the heavy traffic aggravates the food inaccessibility of residents living in fringe areas. However, similar to the results between the KDE method in planar and network spaces, the proportions and distributions of the four patterns related to the walk and metro modes also have similar representations, as shown in Fig 15. In addition, it is easier to see that the great majority of estimated patterns in the urban hub areas are pattern HH, with no risk of being a "food desert" based on all modes in two spaces. In the results of planar method, the clustered but adequate food retails ensure that the residents demand for food will be met. In addition, in the results of the network method, the even distribution of food outlets along the roads ensures widespread accessibility in all modes, except in areas with heavy traffic jams.

As a whole, identification results for all modes based on the planar method underestimate the severity of the "food desert" compared with the results of network method, especially for the bicycle and taxi modes. The simulation of traffic conditions with real-time trajectory data and human activities of accessing food means that the network KDE method under linear tessellation model identifies this phenomenon with a more accurate and finer-grained location. Based on the network-based results in Table 4, we can draw following conclusions. Comparing the proportion between patterns HL and LL and the proportion between patterns HH and LH in all modes, the results from the network method show that residents in two-thirds of low-income areas are more likely to suffer from the "food desert", whereas residents in two-thirds of the high-income areas have a better food-access environment. Judging from the proportion between patterns HH and HL and the proportion between patterns LH and LL, people with

**Table 4. Four patterns with a cross combination of KDE value and income value.**

| pattern name | pattern form | explanation | risk level | proportion (%) | | | | | | | |
|---|---|---|---|---|---|---|---|---|---|---|---|
| | | | | walk | | bike | | taxi | | metro | |
| | | | | planar | network | planar | network | planar | network | planar | network |
| HH | (1,1) | higher accessibility and higher income | 0 | 47.2 | 47.1 | 59.3 | 39.1 | 57.0 | 33.4 | 45.4 | 40.0 |
| HL | (1,0) | higher accessibility and low income | 1 | 15.1 | 16.2 | 31.3 | 12.1 | 27.0 | 10.0 | 13.6 | 11.9 |
| LH | (0,1) | low accessibility and higher income | 2 | 16.1 | 16.1 | 4.0 | 24.1 | 6.2 | 29.8 | 17.9 | 23.2 |
| LL | (0,0) | low accessibility and low income | 3 | 21.7 | 20.6 | 5.4 | 24.7 | 9.8 | 26.8 | 23.1 | 24.9 |

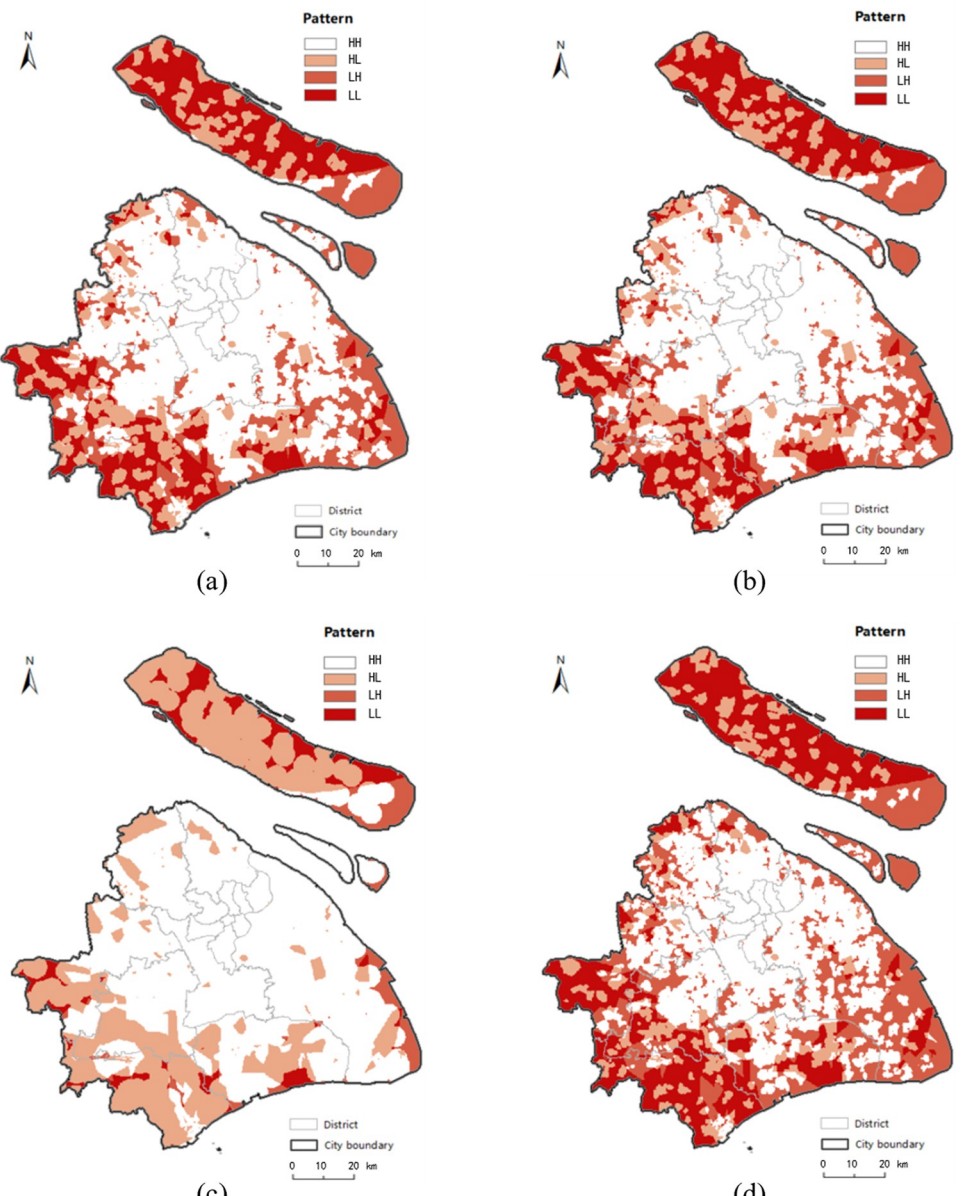

**Fig 14. The distribution of "food desert" patterns in bicycle and taxi mode.** (a) presents the results of the planar KDE analysis in bicycle mode. (b) presents the results of the network KDE analysis in bicycle mode. (c) presents the results of the planar KDE analysis in taxi mode. (d) presents the results of the network KDE analysis in taxi mode. Date Source: Contains information from OpenStreetMap and OpenStreetMap Foundation, which is made available under the Open Database License.

higher accessibility are more possibly to be of higher income; however, the low-income population is slightly greater than the high-income population in low access areas, except for in taxi mode. As found in previous studies in other cities [13, 40], people in disadvantaged areas of Shanghai are more susceptible to negative food-access environments.

Visualizing the above results in Figs 14(B), 14(D) and 15(B), 15(D), we find that "food desert" phenomenon is mostly located in southwest of the city fringe and northwest of Chongming District, which are areas of agricultural land with poor development, but "food deserts" are

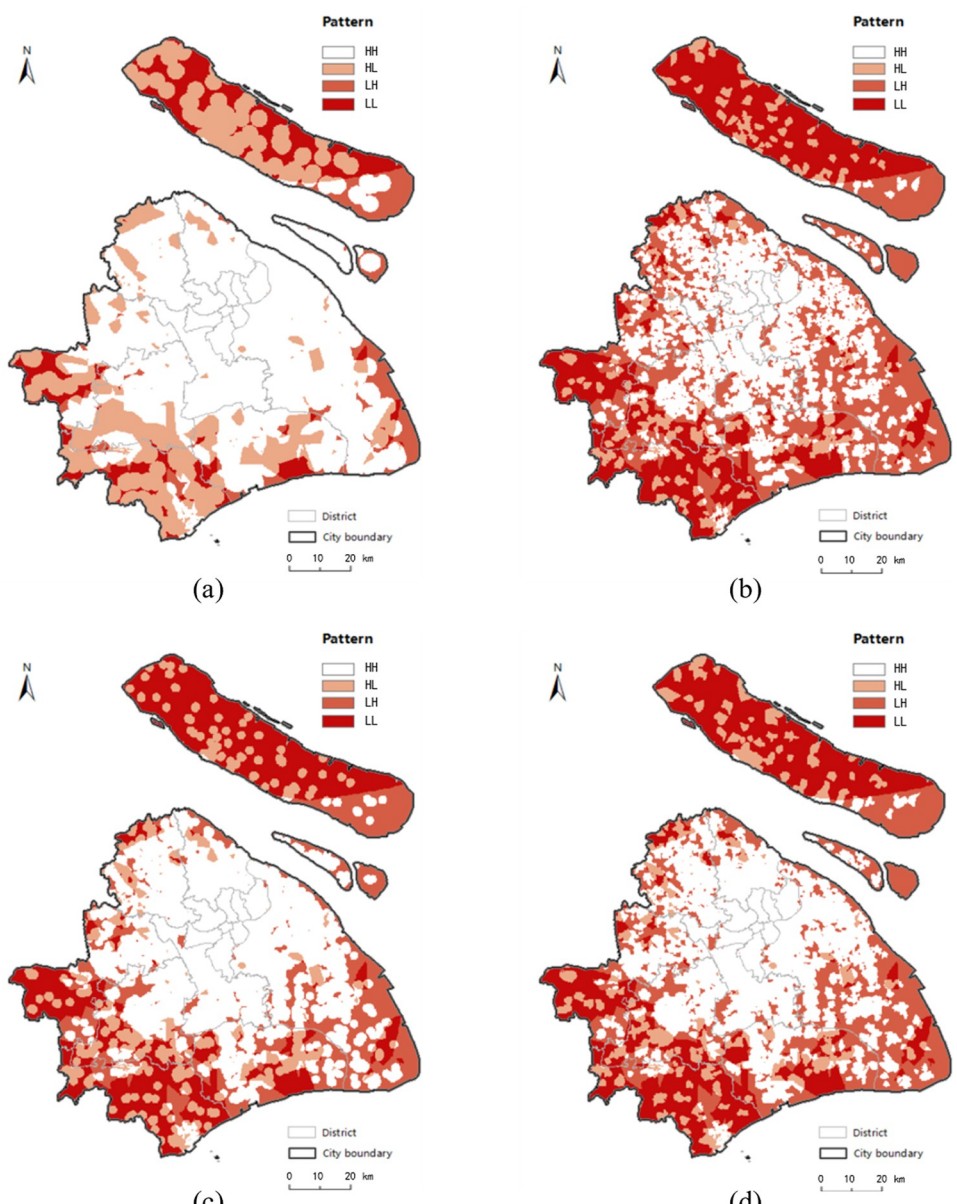

**Fig 15. The distribution of "food desert" patterns in walk and metro mode.** (a) presents the results of the planar KDE analysis in walk mode. (b) presents the results of the network KDE analysis in walk mode. (c) presents the results of the planar KDE analysis in metro mode. (d) presents the results of the network KDE analysis in metro mode. Date Source: Contains information from OpenStreetMap and OpenStreetMap Foundation, which is made available under the Open Database License.

rarely distributed in the developed central city area, which is the core carrier of economy, finance, trade and shopping centres in Shanghai. It is not difficult to understand that inequality of food access may result from the imbalance development between the city fringe and center to some extent. To alleviate this issue, the government is making an effort to balance the development of urban and rural areas, which has published in "Shanghai Comprehensive Master Plan (2017–2035 years)" [66].

Note that the population living in pattern LH area are potentially at risk of this phenomenon, but they are easier to be ignored by public policymakers. Examining the characteristics of the people in this pattern, a fraction of them are located around "food desert" area, the rest are located in the southeast. However, the incomes of most of them are just slightly higher than the low-income standard. In particular, the southeast area located in the tail of "urban development east-west spindle" has a powerful development momentum, which is resulting in rising house price and inflation [47]. As a consequence, people in this area are becoming unable to afford the rising prices. If this trend continues, people in pattern LH will be prone to suffer from the unavailability of affordable healthy food.

## 3.4 Implications for urban planning

As shown, no matter which transport mode citizens choose, there is indeed an imbalance in the food environment between the urban and the suburb within the Shanghai region. Therefore, it is the responsibility of urban planners to develop appropriate planning to help residents address the limited access to affordable healthy foods, especially for the low-income people.

On the one hand, reasonable planning for the distribution and number of full-service healthy food outlets is undoubtedly the more immediate solution in the short term for all transport modes. In particular, the problem of clustered food POIs in the fringe regions needs to be improved by means of setting new food outlets or rearranging existing food outlets. In corresponding planning, the site selection of food outlets is supposed to ensure the uniform distribution of food resources within the fringe regions. In addition, more markets with low prices should be established in low-income residential areas to improve the "food desert" phenomenon.

On the other hand, different plannings of transportation can be made for different areas to relief the phenomenon of food deserts. First, increasing the popularity of subway facilities is a worthwhile approach that has been put on the agenda by urban planners as a planning to alleviate the "food desert" phenomenon in the long run. Second, increasing the number of shared bikes in city fringes is an effective solution to relieve the "food desert" phenomenon. Because the popularity of "bike sharing" and "e-bike sharing" project has benefited many people in China due to its low cost and convenience, which lead people to prefer to select shared bikes when traveling within a short distance [67]. With the expansion of urbanization and private cars, bikeway has been occupied by other transportation modes, especially in urban hubs, which increases the difficulty of performing indispensable activities by bike [68]. To address this issue, local planning departments are suggested to strengthen the construction of "slow traffic facilities", make streets one-way to improve safety for cyclists, build protected bike lanes [69, 70], and establish more bike lanes and docking points along the main trunk traffic road to alleviate traffic pressure so as to enhance the food accessibility in urban center with heavy traffic. For the taxi mode, "odd-and-even license plate rule" has played a role in managing traffic congestion in urban areas [71], it may improve food acquisition to a small degree.

## 4 Conclusions

### 4.1 Summary

This paper proposed a linear tessellation model to simulate activities of food acquisition within road networks and applies it to identify the "food desert" phenomenon of walk, bicycle, taxi and metro mode within a city of developing country, namely Shanghai, China. We developed a network KDE analysis method under a linear tessellation model with real-time trajectory data for four transport modes. The KDE values of the planar method were greater than those of the network method, especially in walk and metro modes, where the KDE value ranges were

7.7–1973.0 and 8.0–2049.2 in the planar results and they were 0.6–567.5 and 2.3–667.5 in the network results. The greatest KDE values of the planar results were nearly three times higher than those of the network results in walk and metro modes. The planar KDE results had larger scope than the network KDE results, which was obvious in bicycle and taxi modes, as evidenced by the fact that the scopes of the planar results with 90.6% and 84.0% were almost twice as large as those of the network results with 50.5% and 47.3%. These differences between two KDE methods under four transport modes indicated that food accessibility was not only related to the distance to food outlets, but also related to the actual travel constraints encountered in the process of acquiring food, such as real-time traffic conditions, road planning and the accessibility of transport facilities. The cumulative number of points in the results of the K-function method within planar space was more than ten times higher than that within network space at the same distance. Taking the K-function results of urban center region for instance, the cumulative number of points at 4.5km was around 250 in the planar method but it was 13 in the network method. According to these differences of results of the K-function method between two spaces within urban centers, urban fringes, suburbs and whole city areas, we analyzed the reasons of unbalanced food accessibility from the perspective of food outlets' distribution pattern. The income data was further overlaid with the estimated KDE value data to identify the "food desert" areas and assess the risk level of other areas with a cross combination of two indicators: low income and low accessibility. As can be seen in the results, 50% of Shanghai was characterized by low food accessibility, and half of these areas were disadvantaged and low-income areas in suburbs, which were the locations experiencing the "food desert" phenomenon. It demonstrated that the food environment was severely lacking equality between urban and suburb areas under all transport modes. Therefore, we provided countermeasures for urban planners to prevent this imbalance from worsening in Shanghai from the planning of food outlets' distribution and transportation facilities.

## 4.2 Innovation points

In contrast to previous network analysis on food accessibility, we divided the road network segments into isochronal linear units based on real-time speed to measure a fine-grained level of accessibility in network space, which translated the analysis scale from distance to time and allowed exploration of diversity within the same road segment. Then, conforming to Tobler's First Law, introducing the KDE method to network space provided a comprehensive density measurement for more than food outlets and road density. The KDE value reflected all possible paths for all stores within the time threshold rather than using minimum, maximum or mean travel time single indicators adopted in traditional studies. The linear tessellation model has offered a reasonable space and flexible scale for analyzing food-access activities constrained by road networks. It would provide an innovative analytical framework that can be applied to other human activities in network space, such as traffic accidents, street crimes and urban function partition.

## 4.3 Limitations and future research

There were several limitations in this study. Firstly, we identified the locations of "food desert" areas (pattern LL) based on the definition and assess the risk of "food desert" in other areas (patterns HH, HL, LH) but did not explore the association between socioeconomic variables and the disparity of food accessibility. Secondly, we ignored the scale among different food stores, the relationship between service capability and population, and the dietary habit of residents. We simply measured food acquisition in terms of travel time cost. Thirdly, we analyzed the "food desert" phenomenon using data from only one time period, lacking data from

different time periods to explore that how an area deteriorates into a food desert over time. Finally, subject to income data acquisition at the non-administrative level, we had to utilize an indicator of the housing price-to-income ratio to deduce the income distribution by extracting the house price data within the whole city, which may result in deviation of defined results for "food desert" areas. Further research would treat population density and the diversity of food outlets as analysis factors weighted in the linear tessellation model to more fully estimate the food accessibility. We will utilize data from more time periods and combine socioeconomic variables and data to analysis the "food desert" phenomenon in more depth.

## Acknowledgments

The authors sincerely appreciate academic editor Ph.D. Mohamed R. Abonazel and all editorial assistant, reviewer Tejwant Singh Brar and other anonymous reviewers for their insightful comments, which significantly improved the quality of this paper.

## Author Contributions

**Conceptualization:** Yakun He.

**Data curation:** Lu Wang, Wenjuan Ye.

**Formal analysis:** Lu Wang.

**Funding acquisition:** Junxiao Zhang.

**Investigation:** Lu Wang.

**Methodology:** Lu Wang, Yakun He.

**Project administration:** Zhonghai Yu, Xin Li.

**Resources:** Lu Wang, Wenjuan Ye.

**Supervision:** Zhonghai Yu, Xin Li.

**Visualization:** Lu Wang.

**Writing – original draft:** Lu Wang, Yakun He.

**Writing – review & editing:** Lu Wang, Yakun He, Hongrui Wang, Yingping Liu, Junxiao Zhang.

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
