## [Decision Letter · Decision Letter 0]

25 Jun 2023

PONE-D-23-15204A linear tessellation model for the identification of “food desert”: A case study of Shanghai, ChinaPLOS ONE

Dear Dr. He,

Thank you for submitting your manuscript to PLOS ONE. After careful consideration, we feel that it has merit but does not fully meet PLOS ONE’s publication criteria as it currently stands. Therefore, we invite you to submit a revised version of the manuscript that addresses the points raised during the review process.

We look forward to receiving your revised manuscript.

Kind regards,

Mohamed R. Abonazel, Ph.D.

Academic Editor

PLOS ONE

4. We note that Figures 1, 2, 6, 9, 10, 11 and 12 in your submission contain [map/satellite] images which may be copyrighted. All PLOS content is published under the Creative Commons Attribution License (CC BY 4.0), which means that the manuscript, images, and Supporting Information files will be freely available online, and any third party is permitted to access, download, copy, distribute, and use these materials in any way, even commercially, with proper attribution. For these reasons, we cannot publish previously copyrighted maps or satellite images created using proprietary data, such as Google software (Google Maps, Street View, and Earth). For more information, see our copyright guidelines: http://journals.plos.org/plosone/s/licenses-and-copyright.

a. You may seek permission from the original copyright holder of Figures 1, 2, 6, 9, 10, 11 and 12 to publish the content specifically under the CC BY 4.0 license. 

Additional Editor Comments:

The authors are requested to make appropriate modifications to this manuscript as suggested by the reviewers.

Reviewers' comments:

Reviewer's Responses to Questions

**Comments to the Author**

1. Is the manuscript technically sound, and do the data support the conclusions?

Reviewer #1: Yes

Reviewer #2: Yes

2. Has the statistical analysis been performed appropriately and rigorously? 

Reviewer #1: Yes

Reviewer #2: Yes

3. Have the authors made all data underlying the findings in their manuscript fully available?

Reviewer #1: Yes

Reviewer #2: Yes

4. Is the manuscript presented in an intelligible fashion and written in standard English?

Reviewer #1: Yes

Reviewer #2: Yes

5. Review Comments to the Author

Reviewer #1: This study evaluates the "food desert" phenomenon in Shanghai using a linear tessellation and network constrained kenel density methods. It is an interesting topic and also an important issue in China that has rarely been concerned. This work presents a certain level of novelty and significance. However, some contents are not very clear and the details should be clarified. Here are my comments, and i hope they would be helpful for improvement:

1. The English is fluent but some expressions are weird to me. For example, in section 2.1, "numbering 12419 in total" could be presented in a simpler way like "We obtained 12,419 records of the food retail outlets via ....". In section 3.3, "average house price square meter" could be "averge house price per square meter". Please ask a native speaker to polish this article and make a throughout check on such issues.

2. Equation 1 seems to be unclear. Firstly, some varibles are repeatly defined in equation 1 and 2. Secondly, what is f_j means in this study? Is this variable equivalent to f^i (speed-based weight). Why this weight is calculated as the ratio between the speed and l_e? In that sense L_i should equal to vi if there is only one f? In the experiment, L_i seems to represent the movement that an object could move in 1s, which is confused to me. Please clarify this.

3. I dont this k((s-c)/tau) is a good expression in equation 4, whilst k(x), x= (s-c)/tau might be more clear.

4. What do the street speed means in metro mode and how does it measures? Because the metro network should be another network rather that the road, it is hard to interpret the results in street-level on metro mode.

5. In section 3.3, the conclusion "the results from the network method show that two-thirds of the low-income residents are more likely to suffer from the “food desert”, whereas two-thirds of the high-income residents have a better food-access environment" could be problematic because i don' think the actual population distribution has been considered in this study. The statistic here is just the propotion of area rather than the real population number.

minor comments:

1. section 2.2.1, please add some reference for Kalman filtering and state the benefit for using it here.

2. The authors may want to add the name of travel mode as the title of every subplot in like figure 10, 11.

Reviewer #2: [Summary of contributions]

The paper presents an in-depth analysis of food accessibility in urban and suburban areas, focusing on the phenomenon of "food deserts". It uses a combination of Kernel Density Estimation (KDE) and spatial overlay analysis to identify areas at risk of becoming a food desert. The paper also provides a comparative analysis of the planar and network methods in estimating the patterns of food accessibility. Further, the paper provides a nuanced understanding of the food desert phenomenon by defining four patterns of risk, which can be useful for policymakers and urban planners.

[Strong points]

S1. An innovative framework for identifying "food desert" areas, which is highly meaningful for urban planning and public health.

S2. Real-world Application with different modes of transport and their impact on food accessibility.

S3. A detailed classification of the "food desert" phenomenon with four patterns of risk, ranging from risk-free areas to top risk level areas, which are the actual "food deserts".

S4. A comparative analysis of the planar and network methods in estimating the patterns of food accessibility.

[Opportunities for improvement]

O1. The results in this paper are mostly statistical and rather abstract. It would improve the paper if the authors add some concrete case studies to demonstrate the effectiveness of the proposed method. In particular, it would be nice if the authors can show examples of different stages of food deserts.

O2. The paper can benefit from analysis on the impact of different government policies or interventions aimed at improving food accessibility and reducing "food desert" phenomena.

O3. The authors apply specifically KDE and spatial overlay analysis. I'm curious if other data analysis methods can be used for the same purpose, such as clustering, graph analytics, and dimensionality reduction methods.

O4. Finally, the study can be enriched by also analyzing the temporal aspects of the data, e.g., a longitudinal study showing how an area deteriorates into a food desert over time, and how different classes of food deserts can gradually transform into other classes.

I suggest that the authors carefully revise the paper to address the above points.

[Minor issues]

D1. The phrase "the quantity and distribution of food outlets satisfy the majority of residents’ demands for food access considering the aspect of distance but without considering the real behaviour of acquiring" could be rephrased for clarity. It's not entirely clear what "the real behaviour of acquiring" refers to.

D2. The sentence "Furthermore, the heavy traffic aggravates their inaccessibility" could be more specific. It's not clear whose inaccessibility is being referred to.

D3. The phrase "the even distribution of food outlets along the roads determines the widespread accessibility in all modes except for the areas with heavy traffic jams" could be rephrased for clarity. It might be clearer to say something like "the even distribution of food outlets along the roads ensures widespread accessibility in all modes, except in areas with heavy traffic jams".

6. PLOS authors have the option to publish the peer review history of their article (what does this mean?). If published, this will include your full peer review and any attached files.

Reviewer #1: No

Reviewer #2: No

---

## [Author Response · Author response to Decision Letter 0]

26 Mar 2024

All the responds to the comments from reviewers and editors have unloaded in the file named "Response to Reviewers.docx". It is hard to show the revised figures and formulas in this box. Here we show the responds simply. Please refer to the file named "Response to Reviewers.docx" for detail responds.

Responses

We thank the editors and reviewers for their valuable requirements and comments, which have enabled us to significantly improve this paper. We carefully revised the manuscripts and corresponding figures. We chose the track changes mode in Word to highlight our changes in the manuscripts, where the texts in blue and strikethrough refers to the deleted contents, and the texts in red and underline refers to the added contents. The original Figures 1 to 12 have been modified and renumbered as Figures 1 to 15.

Please find our responses to each individual requirement and comment below.

Comments from the Editors and Reviewers:

Editors

Requirements to the Author

The responses to all comments are as follows:

Thank you for this requirement. 

We have revised the title page, manuscripts, figures and other submitted files according to PLOS ONE's style requirements. And we uploaded a separate document in “.doc” format containing all the modified figures, and uploaded each revised figure in “.tif” format in the document named “Figures.rar” as Supporting Information.

We hope you are satisfied with all the modifications.

Thank you for this reminder.

We have provided repository information for our data in the form of private link. When our manuscript is accepted, we will change the private link to a public link for readers to browse and download our experimental data. 

We hope you are satisfied with this operation.

Thank you for your requirement.

We have added the separate caption for subgraphs in each figure, as seen in the document named “Figure.doc” and the submitted figures in “.tif” format in the document named “Figures.rar”. 

We hope you are satisfied with the modifications about all the figures.

4. We note that Figures 1, 2, 6, 9, 10, 11 and 12 in your submission contain [map/satellite] images which may be copyrighted. All PLOS content is published under the Creative Commons Attribution License (CC BY 4.0), which means that the manuscript, images, and Supporting Information files will be freely available online, and any third party is permitted to access, download, copy, distribute, and use these materials in any way, even commercially, with proper attribution. For these reasons, we cannot publish previously copyrighted maps or satellite images created using proprietary data, such as Google software (Google Maps, Street View, and Earth). For more information, see our copyright guidelines: http://journals.plos.org/plosone/s/licenses-and-copyright.We require you to either (1) present written permission from the copyright holder to publish these figures specifically under the CC BY 4.0 license, or (2) remove the figures from your submission.

Thank you for this reminder.

In our manuscripts, all the figures are created by Lu Wang who is the first author of this paper. Therefore, she is the copyright holder of Figures 1, 2, 6, 9, 10, 11 and 12. We have sought permission to use the figures 1, 2, 6, 7, 8, 9, 12, 13, 14 and 15 from her. And we uploaded the document named “Permission form.pdf” as a submission file labeled with the file type “Other” in new submission.

5. We note that your author list was updated during the revision process. Please complete our Authorship Change Form by following this link: http://journals.plos.org/plosone/s/file?id=13d0/plos-one-change-to-authorship-form.docx If you are adding or removing more than 2 authors, you can complete multiple forms and submit as many forms as needed to reflect the updates. Please return the form(s) as an attachment by emailing plosone@plos.org or by uploading it as a submission file labeled with the file type ‘Other’. Please note that if your manuscript is accepted, we will not be able to complete the publication process without the completed form.

Thank you for your reminder. 

We have uploaded the document named “Request for change to authorship.pdf” as a submission file labeled with the file type “Other” in new submission. 

6. Please note that PLOS ONE is unable to publish previously copyrighted maps or satellite images, or images created using proprietary data. For these reasons, we cannot publish images generated by software which copyrights their output (such as Google Maps, Street View, and Earth). In order to use these images in your submission, we require explicit permission from the copyright owner to publish the figures under the CC BY 4.0 license.

At this time, please kindly clarify the following regarding Figures 1, 2, 6, 7, 8, 9, 12, 13, 14 and 15:

a) Where did the authors obtain the [maps, satellite images, basemaps, shapefiles, map data, etc.] in Figures 1, 2, 6, 7, 8, 9, 12, 13, 14 and 15?

Thank you for your question.

The data named City boundary, District and Road network in Figures 1, 2, 6, 7, 8, 9, 12, 13, 14 were downloaded from OpenStreetMap (https://www.openstreetmap.org/), which is an open-source platform. We checked the journal's requirements for publication of figures, the data downloaded from the OpenStreetMap conforms to the Creative Commons Attribution (CC BY) license as long as the statement "Contains information from OpenStreetMap and OpenStreetMap Foundation, which is made available under the Open Database License.” (https://journals.plos.org/plosone/s/figures) We have added this statement to the title of the corresponding figures, as shown in the document named “Figure.docx”

The data named “Food outlets” in Figure 1 was obtained via the Baidu API (http://lbsyun.baidu.com/index.php?title=webapi/guide/webservice-placeapi), which is an open-source platform.

The estimated data in Figures 2, 6, 7, 8, 9, 12, 13, 14 and 15 were calculated by the methods of our study.

All the data we used to create maps was is uploaded to figshare at a private link (https://figshare.com/s/267dbe734ba492ac41d6), which has been explained in Data availability statement. When we uploaded data on this website, the licence of data is selected as CC BY 4.0. therefore, our data conforms to CC BY 4.0. When the article is accepted, we will make this link public for everyone to download our data. 

b) Please state whether the [map/satellite images] have been previously copyrighted to your knowledge.

Thank you for this question. 

As we have explained in the question 4, all the figures are created by Lu Wang who is the first author of this paper. Therefore, we have sought permission to use the figures 1, 2, 6, 7, 8, 9, 12, 13, 14 and 15 from her. And we uploaded the document named “Permission form.pdf” as a submission file labeled with the file type “Other” in new submission.

c) If any of the [map/satellite] images have been previously copyrighted, we require specific consent from the copyright holder to publish these images in PLOS ONE, under the CC BY 4.0 license. To seek permission from the copyright owner to publish your map figures under the specific Creative Commons Attribution License (CCAL), CC BY 4.0, please contact them with the following text and PLOS ONE Request for Permission form (https://journals.plos.org/plosone/s/file?id=7c09/content-permission-form.pdf):

Thank you for your reminder. 

All the figures were created by Lu Wang who is the first author of this paper. Therefore, we do not have the [map/satellite] images which have been previously copyrighted.

“I request permission for the open-access journal PLOS ONE to publish the maps in Figures 1, 2, 6, 7, 8, 9, 12, 13, 14 and 15 under the Creative Commons Attribution License (CCAL) CC BY 4.0 (http://creativecommons.org/licenses/by/4.0/). Please be aware that this license allows unrestricted use and distribution, even commercially, by third parties. Please reply and provide explicit written permission to publish these maps under a CC BY license.” Please upload the granted permission to the manuscript as an other file. In the figure caption of the copyrighted figure, please include the following text: “Republished from [ref] under a CC BY license, with permission from [name of publisher], original copyright [original copyright year].”

Thank you for this reminder.

We have uploaded the document named “Permission form.pdf” as a submission file labeled with the file type “Other” in new submission. 

Reviewer 1:

Comments to the Author:

This study evaluates the "food desert" phenomenon in Shanghai using a linear tessellation and network constrained kernel density methods. It is an interesting topic and also an important issue in China that has rarely been concerned. This work presents a certain level of novelty and significance. However, some contents are not very clear and the details should be clarified. Here are my comments, and I hope they would be helpful for improvement.

The responses to all comments are as follows:

Comments:

1. The English is fluent but some expressions are weird to me. For example, in section 2.1, "numbering 12419 in total" could be presented in a simpler way like "We obtained 12,419 records of the food retail outlets via ....". In section 3.3, "average house price square meter" could be "average house price per square meter". Please ask a native speaker to polish this article and make a throughout check on such issues.

Thank you for your corrections.

We have revised the two sentences you mentioned in this comment. The corresponding changes in the manuscripts are as follows:

We obtained 12,419 records of the food retail outlets via the Baidu API (http://lbsyun.baidu.com/index.php?title=webapi/guide/webservice-placeapi), including full-service supermarket chains, fresh markets, sea food stores, fruit and vegetable grocery stores (Su et al., 2017); (Lines 194-198)

Ph is the average house price per square meter; (Lines 526-527)

In addition, our original manuscripts have been edited for proper English language, grammar, punctuation, spelling, and overall style by one or more of the highly qualified native English-speaking editors at AJE which is a professional polish website for papers.

We hope you are satisfied with all the modifications. 

2. Equation 1 seems to be unclear. Firstly, some varibles are repeatly defined in equation 1 and 2. Secondly, what is f_j means in this study? Is this variable equivalent to f^i (speed-based weight). Why this weight is calculated as the ratio between the speed and l_e? In that sense L_i should equal to vi if there is only one f? In the experiment, L_i seems to represent the movement that an object could move in 1s, which is confused to me. Please clarify this.

Thank you for these commons.

For the problem about “some varibles are repeatly defined in equation 1 and 2”, we have rewritten the equations 1 and 2 to help the readers understand them more clearly and deleted the repeated defined varible l_e. (Lines 249-252 and Lines 255-258)

For the second problem, we are sorry that the equation 1 we expressed in the original manuscripts was not clear. We have rewritten the equations 1 and 2 to help the readers understand them more clearly, as shown above.

For the last problem about the definition of the speed-based weight function, the weight calculated by the ratio between the speed and le in order to ensure that the road network is divided into a series of isochronous basic units, where each basic unit represents 1 second. Therefore, Li is equal to vi with this definition of weight function fi, as you mentioned in common. This definition would simply the identification process of “food desert” with the network Kernel Density Estimation method. Because in the process of network kernel density calculation, it is not necessary to calculate the length of the road segments cumulatively, only the number of road segments need to be counted. 

The corresponding explanation about the definition of weight function fi had been mentioned in the original manuscripts, as follows:

 Based on this speed-based weight, the road network is divided into a series of isochronous basic units with varying lengths, where each basic unit represents 1 second. (Lines 263-264)

We hope you are satisfied with all the modifications and explanation.

3. I dont think((s-c)/tau) is a good expression in equation 4, whilst k(x), x= (s-c)/tau might be more clear.

Thank you for your suggestion.

We have modified the expression of equations 3 and 4. (Lines 278-280 and Lines 337-341)

We hope you are satisfied with these modifications about equations 3 and 4.

4. What do the street speed means in metro mode and how does it measures? Because the metro network should be another network rather that the road, it is hard to interpret the results in street-level on metro mode.

Thank you for this common.

As shown in Fig 2(d), the speed of metro network with 14 lines were calculated by the average speed of the corresponding line, the speed of road network utilized the calculated speed of the bicycle mode. Because the metro lines are connected with the road lines. The behavior of residents going to the metro station to take the metro usually occurs in the road network. Therefore, we utilized the speed of bicycle mode to simulate the behavior of residents going to the metro station.

We hope you are satisfied with this explanation.

5. In section 3.3, the conclusion "the results from the network method show that two-thirds of the low-income residents are more likely to suffer from the “food desert”, whereas two-thirds of the high-income residents have a better food-access environment" could be problematic because i don' think the actual population distribution has been considered in this study. The statistic here is just the proportion of area rather than the real population number.

Thank you for this correction.

We are sorry that the description of this conclusion is not accurate. We have modified it.

The corresponding changes in the manuscripts are as follows:

Comparing the proportion between patterns 1 and pattern 3 and the proportion between patterns 0 and pattern 2 in all modes, the results from the network method show that residents in two-thirds of low-income areas are more likely to suffer from the “food desert”, whereas residents in two-thirds of the high-income areas have a better food-access environment. (Lines 656-660)

We hope you are satisfied with this modification.

Minor comments:

1. Section 2.2.1, please add some reference for Kalman filtering and state the benefit for using it here.

Thank you for your suggestion.

We have added the references for Kalman filtering and stated its benefit in the revised manuscripts. The corresponding changes in the manuscripts are as follows:

The average speed of each road segment is calculated by trajectory data with Kalman filtering method (Kobayashi et al., 1995). Because Kalman filter can estimate the position and speed of a vehicle from a series of incomplete and noisy data, which makes it an optimal estimation method and is widely used in the field of automatic driving. (Lines 258-262)

The added reference is as follows: 

Kobayashi, K., Cheok, KC., & Watanabe, K., 1995. Estimation of absolute vehicle speed using fuzzy logic rule-based

---

## [Decision Letter · Decision Letter 1]

3 Jun 2024

PONE-D-23-15204R1A linear tessellation model for the identification of “food desert”: A case study of Shanghai, ChinaPLOS ONE

Dear Dr. He,

Thank you for submitting your manuscript to PLOS ONE. After careful consideration, we feel that it has merit but does not fully meet PLOS ONE’s publication criteria as it currently stands. Therefore, we invite you to submit a revised version of the manuscript that addresses the points raised during the review process.

We look forward to receiving your revised manuscript.

Kind regards,

Mohamed R. Abonazel, Ph.D.

Academic Editor

PLOS ONE

Journal Requirements:

Reviewers' comments:

Reviewer's Responses to Questions

**Comments to the Author**

1. If the authors have adequately addressed your comments raised in a previous round of review and you feel that this manuscript is now acceptable for publication, you may indicate that here to bypass the “Comments to the Author” section, enter your conflict of interest statement in the “Confidential to Editor” section, and submit your "Accept" recommendation.

Reviewer #1: All comments have been addressed

Reviewer #3: All comments have been addressed

2. Is the manuscript technically sound, and do the data support the conclusions?

Reviewer #1: Yes

Reviewer #3: Yes

3. Has the statistical analysis been performed appropriately and rigorously? 

Reviewer #1: Yes

Reviewer #3: Yes

4. Have the authors made all data underlying the findings in their manuscript fully available?

Reviewer #1: Yes

Reviewer #3: Yes

5. Is the manuscript presented in an intelligible fashion and written in standard English?

Reviewer #1: No

Reviewer #3: Yes

6. Review Comments to the Author

Reviewer #1: The author has essentially addressed the issues I raised last time. I believe that the overall technical approach and conclusions of this article are not problematic, but there are still some areas that need improvement in terms of writing. Here are some of my suggestions:

1. In the abstract and on line 65, the author specifically mentions the US. Since the subsequent text does not focus on the issue of food deserts in the US, it would be more appropriate to use “in many countries worldwide” here.

2. On line 80, the term “Density” requires clarification. What density is being referred to? The author needs to be more specific.

3. On line 93, there is no need to quote Professor Tobler's original statement.

4. On line 96, the k-function is introduced. This can be presented as a separate paragraph since it is quite different from the previous contents.

5. On line 148, what does it mean by “current study”? Is there a significant difference between this and a literature review?

6. On line 186, abbreviations like POI should be accompanied by their full name the first time they appear.

7. On line 193, the full name of SODA might be “Shanghai Open Data Apps”, and at least should not include “competition.”

8. From line 192 to 199, the information about these websites can be inserted as footnotes in the document.

9. For Figures 14 and 15, I do not think using “pattern 0-4” is a good idea. Firstly, there is no “pattern 2” here, which seems odd. Moreover, using numbers does not convey the actual meaning of the patterns. You can refer to the commonly used expressions in GIS, such as LL (low-low), to express the patterns you are interested in.

10. There are too many references in this article, and some of them are ever redundant. For example, it might not be necessary to cite ten references to demonstrate the importance of food accessibilty to low income population without reviewing in their details.

11. Figure 3, please keep the font and style consistent in figures. For exmaple, some words are relatively too small in figure 3 and the first-letter vary between uppercase and lowercase. Similar issues also exist in Figure 4. it will be better if "the result of..." could be "The result of ...", "projection point" could be "Projection point"....

12. The unit for distance measurement should be consistent throughout the manuscipt. In Figure 11, the distance unit is meter, while in figure 13, for instance, the unit is mile.

Reviewer #3: Conclusions Should be supported by data, Kindly add data to support the conclusions with KDE values and K Function Curves.

No References should be added in conclusion section so remove all in text references from conclusion section.

7. PLOS authors have the option to publish the peer review history of their article (what does this mean?). If published, this will include your full peer review and any attached files.

Reviewer #1: No

Reviewer #3: **Yes: **Tejwant Singh Brar

---

## [Author Response · Author response to Decision Letter 1]

7 Jul 2024

Responses

We thank the reviewers for their valuable comments, which have enabled us to significantly improve this paper. We carefully revised the manuscripts and corresponding figures. We chose the track changes mode in Word to highlight our changes in the manuscripts, where the texts in blue and strikethrough refers to the deleted contents, and the texts in red and underline refers to the added contents. The line number mentioned is the revised manuscript without changes marked.

Please find our responses to each individual comment below.

Journal Requirements:

Thank you for your requirement. 

According to a comment from reviewer 1, he/she thinks there are too many references in our paper, and some of them are ever redundant. Therefore, we remove some redundant references, and the number of references is reduced from 90 to 72 in the revised manuscripts.

The removed 18 references are as follows:

[1] Apparicio, P., Cloutier, M. & Shearmur, R., 2007. The case of Montréal's missing food deserts: Evaluation of accessibility to food supermarkets. Int. J. Health Geogr. 6(4), 1-13.

[2] Bailey, T. C., Gatrell, A.C., 1995. Interactive spatial data analysis. New York, NY: John Wiley and Sons.

[3] Bailey, T.C., Gatrell, A.C., 1995. Interactive Spatial Data Analysis. Longman Scientific, Harlow, UK.

[4] Basu, S., & Vasudevan, V., 2013. Effect of bicycle friendly roadway infrastructure on bicycling activities in urban India. Part of Special Issue: 2nd Conference of Transportation Research Group of India. P. Chakroborty, H.K. Reddy and A. Amekudzi (eds.). Procedia – Social and Behavioral Sciences. 104, 1139-1148.

[5] Bell, J., Mora, G., Hagan, E., et al., 2013. Access to healthy food and why it matters: A review of the research. Philadelphia, PA: The Food Trust.

[6] Bureau, S. S., 2018. Shanghai statistical bulletin on national economic and social development. Statistical science and practice. 03, 11-21.

[7] Cummins, S., & Macintyre, S., 2002. “Food deserts”—evidence and assumption in health policy making. BMJ. 325(7361), 436-438.

[8] Cummins, S., 2014. Food deserts. The Wiley Blackwell Encyclopedia of Health, Illness, Behavior, and Society.

[9] Epanechnikov, V. A., 1969. Non-parametric estimation of a multivariate probability density. Theory Probab. Appl. 14(1), 153-158.

[10] Kobayashi, K., Cheok, KC., & Watanabe, K., 1995. Estimation of absolute vehicle speed using fuzzy logic rule-based Kalman filter. In: Proceedings of 1995 American Control Conference-ACC'95. IEEE, Vol. 5, pp. 3086-3090.

[11] Nie, K., Wang, Z., Du, Q., et al., 2015. A network-constrained integrated method for detecting spatial cluster and risk location of traffic crash: A case study from Wuhan, China. Sustainability. 7(3), 2662-2677.

[12] Rhone, A., Ver Ploeg, M., Dicken, C., et al., 2017. Low-income and Low-supermarket-access Census Tracts, 2010-2015. Washington, DC: US Department of Agriculture, Economic Research Service.

[13] Silverman, B.W., 1986. Density Estimation for Statistics and Data Analysis. Chapman Hall, London, UK.

[14] Wang, M., 2018. Exploring the direction of “city-oriented policy” in key cities from the perspective of “price-to-income ratio deviation”. Shanghai Real Estate. 376(06):14-17.

[15] Wang, S., Zhang, J., Liu, L., et al., 2010. Bike-Sharing-A new public transportation mode: State of the practice & prospects. In: Emergency Management and Management Sciences (ICEMMS), 2010 IEEE International Conference on, pp. 222e225

[16] Wang, S., Zhang, J., Liu, L., et al., 2010. Bike-Sharing-A new public transportation mode: State of the practice & prospects. In: Proceedings of 2010 IEEE International Conference on Emergency Management and Management Sciences, ICEMMS 2010, Beijing, China, August 8, 2010 - August 10, 2010, pp. 222-225.

[17] Wrigley, N., Warm, D., & Margetts, B., 2003. Deprivation, diet, and food-retail access: Findings from the Leeds ‘food deserts' study. Environ. Plan. A. 35(1), 151-188.

[18] Xie, Z., Yan, J., 2013. Detecting traffic accident clusters with network kernel density estimation and local spatial statistics: an integrated approach. J. Transp. Geogr. 31, 64-71.

[19] Zhang, T. W., 2002. Urban development and a socialist pro-growth coalition in shanghai. Urban Aff. Rev. 37(4), 475-499.

In addition, we have completed and corrected all the references. And we numbered the references in the order in which they were cited and inserted the numbers in the corresponding places.

We hope you are satisfied with all the modifications of references.

Comments from Reviewers:

Review Comments to the Author

The responses to all comments are as follows:

Reviewer 1:

Comments to the Author:

The author has essentially addressed the issues I raised last time. I believe that the overall technical approach and conclusions of this article are not problematic, but there are still some areas that need improvement in terms of writing.

The responses to all comments are as follows:

Comments:

1. In the abstract and on line 65, the author specifically mentions the US. Since the subsequent text does not focus on the issue of food deserts in the US, it would be more appropriate to use “in many countries worldwide” here.

Thank you for your corrections.

We have changed “in the United States and other countries” to “in many countries worldwide” in the revised manuscripts. (Lines 63-64)

2. On line 80, the term “Density” requires clarification. What density is being referred to? The author needs to be more specific.

Thank you for this suggestion.

We have a more specific explanation of density-based methods in the revised manuscripts. The corresponding changes are as follows:

The density-based methods quantify the accessibility of food outlets with total count, count per population or count per square area using the buffer method, kernel density estimation (KDE) or spatial clustering. (Lines 75-77)

We hope you are satisfied with all the modifications and explanation.

3. On line 93, there is no need to quote Professor Tobler's original statement.

Thank you for your suggestion.

We have deleted the Professor Tobler's original statement of Tobler's First Law in the revised manuscripts. (Line 86)

4. On line 96, the k-function is introduced. This can be presented as a separate paragraph since it is quite different from the previous contents.

Thank you for this common.

According to your suggestion, the introduction of the k-function has been as a separate paragraph in the revised manuscripts. (Line 90)

5. On line 148, what does it mean by “current study”? Is there a significant difference between this and a literature review?

Thank you for this question.

The section 1.3 Current study refer to our study objective of this paper. To make this section clearer for readers, we changed the title to “Study objective” in the revised manuscripts. (Line 136)

6. On line 186, abbreviations like POI should be accompanied by their full name the first time they appear.

Thank you for this correction.

We have added the full name of POIs in the revised manuscripts. (Line 171)

7. On line 193, the full name of SODA might be “Shanghai Open Data Apps”, and at least should not include “competition.”

Thank you for this correction.

We have corrected the full name of SODA as “Shanghai Open Data Apps” in the revised manuscripts. (Line 178)

8. From line 192 to 199, the information about these websites can be inserted as footnotes in the document.

Thank you for your suggestion.

The corresponding the URL has been inserted as footnotes in the revised manuscripts. (Lines 167-182)

9. For Figures 14 and 15, I do not think using “pattern 0-4” is a good idea. Firstly, there is no “pattern 2” here, which seems odd. Moreover, using numbers does not convey the actual meaning of the patterns. You can refer to the commonly used expressions in GIS, such as LL (low-low), to express the patterns you are interested in.

Thank you for this comment.

According to your suggestion, we added the pattern name to express four patterns. Pattern name HH expresses pattern of the higher accessibility and higher income. Pattern name HL expresses pattern of the higher accessibility and lower income. Pattern name LH expresses pattern of the low accessibility and higher income. Pattern name LL expresses pattern of the low accessibility and low income. The original pattern index is changed to risk level in the revised manuscripts and Table 3. We have added explanation of the corresponding modification. The changes are as follows:

Finally, we can obtain four patterns with cross combination, as shown in Table 4. The four patterns are named as HH, HL, LH and LL, corresponding to the risk level 0, 1, 2 and 3, respectively, where the higher risk level represents the higher risk of “food desert”. (Lines 494-496) 

In addition, we have changed patterns 0, 1, 2 and 3 to patterns HH, HL, LH and LL, respectively, in the revised manuscripts and Figures 14 and 15.

We hope you are satisfied with all the modifications.

10. There are too many references in this article, and some of them are ever redundant. For example, it might not be necessary to cite ten references to demonstrate the importance of food accessibilty to low income population without reviewing in their details.

Thank you for this suggestion.

We have removed the redundant references. The number of references has been reduced from 90 to 72 in the revised manuscripts.

The removed 18 references are as follows:

[1] Apparicio, P., Cloutier, M. & Shearmur, R., 2007. The case of Montréal's missing food deserts: Evaluation of accessibility to food supermarkets. Int. J. Health Geogr. 6(4), 1-13.

[2] Bailey, T. C., Gatrell, A.C., 1995. Interactive spatial data analysis. New York, NY: John Wiley and Sons.

[3] Bailey, T.C., Gatrell, A.C., 1995. Interactive Spatial Data Analysis. Longman Scientific, Harlow, UK.

[4] Basu, S., & Vasudevan, V., 2013. Effect of bicycle friendly roadway infrastructure on bicycling activities in urban India. Part of Special Issue: 2nd Conference of Transportation Research Group of India. P. Chakroborty, H.K. Reddy and A. Amekudzi (eds.). Procedia – Social and Behavioral Sciences. 104, 1139-1148.

[5] Bell, J., Mora, G., Hagan, E., et al., 2013. Access to healthy food and why it matters: A review of the research. Philadelphia, PA: The Food Trust.

[6] Cummins, S., & Macintyre, S., 2002. “Food deserts”—evidence and assumption in health policy making. BMJ. 325(7361), 436-438.

[7] Cummins, S., 2014. Food deserts. The Wiley Blackwell Encyclopedia of Health, Illness, Behavior, and Society.

[8] Epanechnikov, V. A., 1969. Non-parametric estimation of a multivariate probability density. Theory Probab. Appl. 14(1), 153-158.

[9] Kobayashi, K., Cheok, KC., & Watanabe, K., 1995. Estimation of absolute vehicle speed using fuzzy logic rule-based Kalman filter. In: Proceedings of 1995 American Control Conference-ACC'95. IEEE, Vol. 5, pp. 3086-3090.

[10] Nie, K., Wang, Z., Du, Q., et al., 2015. A network-constrained integrated method for detecting spatial cluster and risk location of traffic crash: A case study from Wuhan, China. Sustainability. 7(3), 2662-2677.

[11] Rhone, A., Ver Ploeg, M., Dicken, C., et al., 2017. Low-income and Low-supermarket-access Census Tracts, 2010-2015. Washington, DC: US Department of Agriculture, Economic Research Service.

[12] Silverman, B.W., 1986. Density Estimation for Statistics and Data Analysis. Chapman Hall, London, UK.

[13] Wang, M., 2018. Exploring the direction of “city-oriented policy” in key cities from the perspective of “price-to-income ratio deviation”. Shanghai Real Estate. 376(06):14-17.

[14] Wang, S., Zhang, J., Liu, L., et al., 2010. Bike-Sharing-A new public transportation mode: State of the practice & prospects. In: Emergency Management and Management Sciences (ICEMMS), 2010 IEEE International Conference on, pp. 222e225

[15] Wang, S., Zhang, J., Liu, L., et al., 2010. Bike-Sharing-A new public transportation mode: State of the practice & prospects. In: Proceedings of 2010 IEEE International Conference on Emergency Management and Management Sciences, ICEMMS 2010, Beijing, China, August 8, 2010 - August 10, 2010, pp. 222-225.

[16] Wrigley, N., Warm, D., & Margetts, B., 2003. Deprivation, diet, and food-retail access: Findings from the Leeds ‘food deserts' study. Environ. Plan. A. 35(1), 151-188.

[17] Xie, Z., Yan, J., 2013. Detecting traffic accident clusters with network kernel density estimation and local spatial statistics: an integrated approach. J. Transp. Geogr. 31, 64-71.

[18] Zhang, T. W., 2002. Urban development and a socialist pro-growth coalition in shanghai. Urban Aff. Rev. 37(4), 475-499.

[19] Zhang, T. W., 2002. Urban development and a socialist pro-growth coalition in shanghai. Urban Aff. Rev. 37(4), 475-499.

We hope you are satisfied with our modifications.

11. Figure 3, please keep the font and style consistent in figures. For example, some words are relatively too small in figure 3 and the first-letter vary between uppercase and lowercase. Similar issues also exist in Figure 4. it will be better if "the result of..." could be "The result of ...", "projection point" could be "Projection point"....

Thank you for this correction.

To keep the font and style consistent in figures, the first letter of words in Figures 3 and 4 has been changed to uppercase. The subtitles of each figure are lowercase. In addition, the words in Figure 3 have been enlarged. All the changes are as seen in the document named “Figure.doc” and the submitted figures in the document named “Figures.rar”

We hope you are satisfied with these modifications.

12. The unit for distance measurement should be consistent throughout the manuscipt. In Figure 11, the distance unit is meter, while in figure 13, for instance, the unit is mile.

Thank you for this comment.

We have standardized the distance measurement to kilometer throughout all the figures, as seen in the document named “Figure.doc” and the submitted figures in the document named “Figures.rar”. 

We hope you are satisfied with all modifications of figures.

Reviewer 3:

Comments to the Author:

1. Conclusions Should be supported by data, kindly add data to support the conclusions with KDE values and K Function Curves.

Thank you for your meaningful suggestion.

We have added the data to describe the differences of KDE values and K-function results between planer and network space in section Conclusion to support findings of our study. The corresponding changes are as follows:

The KDE values of the planar method are greater than those of the network method, especially in walk and metro modes, where the values of the planar results are nearly three times higher than those of the network results. The planar KDE results have larger scope than the network KDE results, which is obvious in bicycle and taxi modes, as evidenced by the fact that the scopes of the planar results are almost twice as large as those of the network results. These differences between two KDE methods under four transport modes indicate that food accessibility is not only related to the distance to food outlets, but also related to the actual travel constraints encountered in the process of acquiring food, such as real-time traffic conditions, road planning and the accessibility of transport facilities. The cumulative number of points in the results of the K-function method within planar space

---

## [Decision Letter · Decision Letter 2]

29 Jul 2024

PONE-D-23-15204R2A linear tessellation model for the identification of “food desert”: A case study of Shanghai, ChinaPLOS ONE

Dear Dr. He,

Thank you for submitting your manuscript to PLOS ONE. After careful consideration, we feel that it has merit but does not fully meet PLOS ONE’s publication criteria as it currently stands. Therefore, we invite you to submit a revised version of the manuscript that addresses the points raised during the review process.

We look forward to receiving your revised manuscript.

Kind regards,

Mohamed R. Abonazel, Ph.D.

Academic Editor

PLOS ONE

Journal Requirements:

Reviewers' comments:

Reviewer's Responses to Questions

**Comments to the Author**

1. If the authors have adequately addressed your comments raised in a previous round of review and you feel that this manuscript is now acceptable for publication, you may indicate that here to bypass the “Comments to the Author” section, enter your conflict of interest statement in the “Confidential to Editor” section, and submit your "Accept" recommendation.

Reviewer #1: All comments have been addressed

Reviewer #3: (No Response)

2. Is the manuscript technically sound, and do the data support the conclusions?

Reviewer #1: Yes

Reviewer #3: No

3. Has the statistical analysis been performed appropriately and rigorously? 

Reviewer #1: Yes

Reviewer #3: Yes

4. Have the authors made all data underlying the findings in their manuscript fully available?

Reviewer #1: No

Reviewer #3: Yes

5. Is the manuscript presented in an intelligible fashion and written in standard English?

Reviewer #1: Yes

Reviewer #3: No

6. Review Comments to the Author

Reviewer #1: (No Response)

Reviewer #3: 1. Title of Section 1.3 Should be Objectives of the Study and it should be followed by section on Limitation of the study which now is part of conclusions section

2. Line 635-642 in Conclusion is part of introduction and is repetition in summary

3. Write conclusion in present perfect tense (presently written in present tense).

4. Conclusions should be supported by data from analysis, write values to support three times in line 645, and ten times line 653

5. Innovation points Section Should also be written in Present perfect tense.

7. PLOS authors have the option to publish the peer review history of their article (what does this mean?). If published, this will include your full peer review and any attached files.

Reviewer #1: No

Reviewer #3: **Yes: **Dr. Tejwant Singh Brar

---

## [Author Response · Author response to Decision Letter 2]

4 Aug 2024

Responses

We thank the reviewers for their valuable comments, which have enabled us to significantly improve this paper. We carefully revised the manuscripts. We chose the track changes mode in Word to highlight our changes in the manuscripts, where the texts in blue and strikethrough refers to the deleted contents, and the texts in red and underline refers to the added contents. The line number mentioned is the revised manuscript without changes marked.

Please find our responses to each individual comment below.

Journal Requirements:

1.Please review your reference list to ensure that it is complete and correct. If you have cited papers that have been retracted, please include the rationale for doing so in the manuscript text, or remove these references and replace them with relevant current references. Any changes to the reference list should be mentioned in the rebuttal letter that accompanies your revised manuscript. If you need to cite a retracted article, indicate the article’s retracted status in the References list and also include a citation and full reference for the retraction notice.

Thank you for your notice. 

According to comments from reviewer in the last revision, we have removed some redundant references and checked all the remained references to ensure the completeness and accuracy of references.

Thank you for your requirement. 

The link showed in Data availability statement was dead. We have shared our data via Github. The new link of our data is public for everyone, which is https://github.com/wl7/A-linear-tessellation-model-for-the-identification-of-food-desert-A-case-study-of-Shanghai-China. 

Comments from Reviewer 3:

The responses to all comments are as follows:

Comments:

1. Title of Section 1.3 Should be Objectives of the Study and it should be followed by section on Limitation of the study which now is part of conclusions section

Thank you for your corrections.

Section 1.3 was intended to introduce the purpose of this study to the readers. We feel sorry about the inappropriate descriptions of this section. To make it clearer for readers to understand our study purpose and contents, we revised the texts in the manuscripts. The corresponding changes are as follows:

Based on the proposed linear tessellation model, this study attempts to: (1) introducing the network Kernel Density Estimation analysis method into the linear tessellation model to measure transit-varying food accessibility in Shanghai with trajectory data; (2) applying the network K-function method to analyze the estimated food accessibility from the perspective of food outlets’ distribution pattern; (3) identifying the "food desert" areas by the food accessibility and income and discussing the implications for urban planning The proposed model aims to provide an innovative thought with a rational analysis space for food acquirement issue, and to provide reasonable implications for public policymakers by exploring the state of “food desert” at a finer size. (Lines 147-154)

We hope you are satisfied with these modifications.

2. Line 635-642 in Conclusion is part of introduction and is repetition in summary

Thank you for your comment.

We have deleted the part of the conclusion that duplicates the text in summary. And the corresponding changes in the revised manuscripts are as follows:

This paper proposed a linear tessellation model to simulate activities of food acquisition within road networks and applies it to identify the “food desert” phenomenon of walk, bicycle, taxi and metro mode within a city of developing country, namely Shanghai, China. (Lines 596-598)

We hope you are satisfied with these modifications.

3. Write conclusion in present perfect tense (presently written in present tense).

Thank you for your correction.

We have revised the tense of section Conclusion according to the description of each sentence. For example, the sentences describing research results are in past tense or in present perfect tense, the sentences describing research significance are in present tense, the sentences describing research prospect are in future tense. 

We hope you are satisfied with these modifications.

4. Conclusions should be supported by data from analysis, write values to support three times in line 645, and ten times line 653

Thank you for your suggestion.

We have added the experimental data to support three times and ten times in the conclusion. The corresponding are as follows:

The KDE values of the planar method were greater than those of the network method, especially in walk and metro modes, where the KDE value ranges were 7.7-1973.0 and 8.0-2049.2 in the planar results and they were 0.6-567.5 and 2.3-667.5 in the network results. The greatest KDE values of the planar results were nearly three times higher than those of the network results in walk and metro modes. (Lines 600-604)

The planar KDE results had larger scope than the network KDE results, which was obvious in bicycle and taxi modes, as evidenced by the fact that the scopes of the planar results with 90.6% and 84.0% were almost twice as large as those of the network results with 50.5% and 47.3%. (Lines 604-607)

The cumulative number of points in the results of the K-function method within planar space was more than ten times higher than that within network space at the same distance. Taking the K-function results of urban center region for instance, the cumulative number of points at 4.5km was around 250 in the planar method but it was 13 in the network method. (Lines 610-614)

We hope you are satisfied with these modifications.

5. Innovation points Section Should also be written in Present perfect tense.

The responses are same as question 3.

---

## [Decision Letter · Decision Letter 3]

15 Sep 2024

PONE-D-23-15204R3A linear tessellation model for the identification of “food desert”: A case study of Shanghai, ChinaPLOS ONE

Dear Dr. He,

Thank you for submitting your manuscript to PLOS ONE. After careful consideration, we feel that it has merit but does not fully meet PLOS ONE’s publication criteria as it currently stands. Therefore, we invite you to submit a revised version of the manuscript that addresses the points raised during the review process.

We look forward to receiving your revised manuscript.

Kind regards,

Mohamed R. Abonazel, Ph.D.

Academic Editor

PLOS ONE

Journal Requirements:

Reviewers' comments:

Reviewer's Responses to Questions

**Comments to the Author**

1. If the authors have adequately addressed your comments raised in a previous round of review and you feel that this manuscript is now acceptable for publication, you may indicate that here to bypass the “Comments to the Author” section, enter your conflict of interest statement in the “Confidential to Editor” section, and submit your "Accept" recommendation.

Reviewer #1: (No Response)

Reviewer #3: All comments have been addressed

2. Is the manuscript technically sound, and do the data support the conclusions?

Reviewer #1: (No Response)

Reviewer #3: Yes

3. Has the statistical analysis been performed appropriately and rigorously? 

Reviewer #1: (No Response)

Reviewer #3: Yes

4. Have the authors made all data underlying the findings in their manuscript fully available?

Reviewer #1: (No Response)

Reviewer #3: Yes

5. Is the manuscript presented in an intelligible fashion and written in standard English?

Reviewer #1: (No Response)

Reviewer #3: Yes

6. Review Comments to the Author

Reviewer #1: (No Response)

Reviewer #3: 1. Add Limitations Section after objectives as communicated earlier.

2. Revise the section on Objectives as required

7. PLOS authors have the option to publish the peer review history of their article (what does this mean?). If published, this will include your full peer review and any attached files.

Reviewer #1: No

Reviewer #3: **Yes: **Tejwant Singh Brar

---

## [Author Response · Author response to Decision Letter 3]

22 Oct 2024

Responses

We thank the reviewers for their valuable comments, which have enabled us to significantly improve this paper. We carefully revised the manuscripts. We chose the track changes mode in Word to highlight our changes in the manuscripts, where the texts in blue and strikethrough refers to the deleted contents, and the texts in red and underline refers to the added contents. The line number mentioned is the revised manuscript without changes marked.

Please find our responses to each individual comment below.

Journal Requirements:

Thank you for your notice. 

According to comments from reviewer in the last revision, we have removed some redundant references. This time, we checked all the remained references and revised references 1, 2, 8, 42, 54, 58, 59, 62, 67 to make them complete and correct.

We hope you are satisfied with these modifications.

Comments from Reviewer 3:

The responses to all comments are as follows:

Comments:

1. Add Limitations Section after objectives as communicated earlier.

Thank you for this comment.

The comment you made in the last revision is “Title of Section 1.3 Should be Objectives of the Study and it should be followed by section on Limitation of the study which now is part of conclusions section”. 

In our last response, we have revised the context of Section 1.3 as follows:

Based on the proposed linear tessellation model, this study attempts to: (1) introducing the network Kernel Density Estimation analysis method into the linear tessellation model to measure transit-varying food accessibility in Shanghai with trajectory data; (2) applying the network K-function method to analyze the estimated food accessibility from the perspective of food outlets’ distribution pattern; (3) identifying the "food desert" areas by the food accessibility and income and discussing the implications for urban planning The proposed model aims to provide an innovative thought with a rational analysis space for food acquirement issue, and to provide reasonable implications for public policymakers by exploring the state of “food desert” at a finer size. 

We apologize for not explaining this revision well enough in our last response. In Section 1.2 Literature review, we summarized the problems that exist in the current researches on the analysis of food deserts phenomenon. Therefore, in Section 1.3, we present that our study intends to propose a linear tessellation model to address the problems summarized in Section 1.2, and we simply listed three study objectives to be achieved based on the proposed linear tessellation model. The context of subsequent Section 2 and 3 mainly focused on the linear model and three study objectives. 

The Section 4.3 Limitations you mentioned in this comment is summarized based on the experimental results of this study and provides directions for future research. If Section 4.3 Limitations is placed after Section 1.3 Study objectives, readers may be confused about Section 4.3 because they have not yet understood the methods and experiments. 

Therefore, in our last revision, we modified the contents of Section 1.3, and Limitations wad still placed in the Section 4 conclusions, which aims to make the article logic smoother. To more accurately express the content of section 4.3, the title of this section is changed as “Limitation and future researches”.

We hope you are satisfied with our explanation.

2. Revise the section on Objectives as required.

Thank you for this comment.

In our last response, we have revised Section 1.3 Study objective to make it clearer for readers to understand our study purpose and contents. The corresponding changes are reflected in the response to question 1.

---

## [Decision Letter · Decision Letter 4]

11 Nov 2024

PONE-D-23-15204R4A linear tessellation model for the identification of “food desert”: A case study of Shanghai, ChinaPLOS ONE

Dear Dr. He,

Thank you for submitting your manuscript to PLOS ONE. After careful consideration, we feel that it has merit but does not fully meet PLOS ONE’s publication criteria as it currently stands. Therefore, we invite you to submit a revised version of the manuscript that addresses the points raised during the review process.

We look forward to receiving your revised manuscript.

Kind regards,

Mohamed R. Abonazel, Ph.D.

Academic Editor

PLOS ONE

**Journal Requirements:**

Reviewers' comments:

Reviewer's Responses to Questions

**Comments to the Author**

1. If the authors have adequately addressed your comments raised in a previous round of review and you feel that this manuscript is now acceptable for publication, you may indicate that here to bypass the “Comments to the Author” section, enter your conflict of interest statement in the “Confidential to Editor” section, and submit your "Accept" recommendation.

Reviewer #3: (No Response)

2. Is the manuscript technically sound, and do the data support the conclusions?

Reviewer #3: Yes

3. Has the statistical analysis been performed appropriately and rigorously? 

Reviewer #3: Yes

4. Have the authors made all data underlying the findings in their manuscript fully available?

Reviewer #3: Yes

5. Is the manuscript presented in an intelligible fashion and written in standard English?

Reviewer #3: Yes

6. Review Comments to the Author

**Reviewer #3: **1. Check the grammar in the title of Section 4.3, Its title should be Limitations and Future Research.

2. Remove References from Objectives Section they should be part of literature review section and objectives should be framed by you based on the research gap identified after the literature review.

3. Paragraph one in Section 1.3 should be part of literature review and paragraph 2 of section should only be part of objectives section.

7. PLOS authors have the option to publish the peer review history of their article (what does this mean?). If published, this will include your full peer review and any attached files.

Reviewer #3: **Yes: **Tejwant Singh Brar

---

## [Author Response · Author response to Decision Letter 4]

1 Dec 2024

Responses

We thank the reviewers for their valuable comments, which have enabled us to significantly improve this paper. We carefully revised the manuscripts. We chose the track changes mode in Word to highlight our changes in the manuscripts, where the texts in blue and strikethrough refers to the deleted contents, and the texts in red and underline refers to the added contents. The line number mentioned is the revised manuscript without changes marked.

Please find our responses to each individual comment below.

Journal Requirements:

Thank you for your notice. 

We have corrected the references in the last revision.

2. Please ensure that you include a legends for Figure 1 and 2 within your main document. We do appreciate that you have a figure legends document uploaded as a separate file, however, we do require this to be part of the manuscript file itself and not uploaded separately. 

Thank you for your suggestion. 

We have added the legends of Figs 1 and 2 in our revised manuscripts. 

The changes are as follows: (Lines 161-185)

Shanghai’s rapid urbanization and expansion have been affecting the communication, travel, work and lifestyles of millions of residents. It displays the positive and negative outcomes of urban growth, which makes it a typical place for diverse urban research [57]. 

To apply the proposed method to identify the food accessibility in the Shanghai region as shown in Fig 1(a), where the urban center and urban fringe area is within the red line and blue line, respectively, the remaining area is suburb. 

The study data are as follows: 

(1) Healthy food retail outlets. We obtained 12,419 records of the food retail outlets via the Baidu API, including full-service supermarket chains, fresh markets, sea food stores, fruit and vegetable grocery stores [13], as shown in the green points in Fig 1(b);

(2) Entire road networks. These data are downloaded from OpenStreetMap, as shown in the grey lines in Fig 1(b); 

(3) House price Point of Interests (POIs). These data are obtained from the houses put up for sale in the Lianjia website in March, 2017. The total number of points is 91,619.

(4) Trajectory data. Allowing for the travel mode of residents and low ownership rate for private cars, four types of trajectory data, walk, bicycle, metro and taxi, are selected to calculate the average speed of each road segment for the corresponding travel mode. The estimated speed of the walk, bicycle, taxi and metro mode is shown in Fig 2(a), (b), (c) and (d), respectively.

GPS trajectory data of walk and bicycle are obtained from Yidong GPS APP. GPS trajectory data for taxi are obtained from SODA (Shanghai Open Data Apps). All the GPS trajectory data are in the same time period (March 1 to 31, 2017). The speed of metro mode is set based on the average speed of 14 metro lines in Shanghai, as shown in the illustration in the lower right corner of Fig 1(b). The average speed of 14 metro lines is calculated according to the total metro lines’ length provided by the Shanghai Survey and Mapping Institute and the total metro operation time provided by the Shanghai Metro official website.

3. Can you please upload an additional copy of your revised manuscript that does not contain any tracked changes or highlighting as your main article file. This will be used in the production process if your manuscript is accepted. Please amend the file type for the file showing your changes to Revised Manuscript w/tracked changes. Please follow this link for more information: http://blogs.PLOS.org/everyone/2011/05/10/how-to-submit-your-revised-manuscript/

Thank you for this notice.

We have uploaded the revised manuscript without tracked changes in the submission system.

4. Please ensure that you include a legends/captions for Figure 1 and 15 within your main document.

Thank you for this requirement.

We have added all the figures and their captions in the main document. 

Comments from Reviewer 3:

The responses to all comments are as follows:

Comments:

1. Check the grammar in the title of Section 4.3, Its title should be Limitations and Future Research.

Thank you for your correction.

We have changed the title of Section 4.3 as “Limitations and Future Research” in the revised manuscripts. (Line 655)

We hope you are satisfied with our modification.

2. Remove References from Objectives Section they should be part of literature review section and objectives should be framed by you based on the research gap identified after the literature review.

Thank you for this comment.

We have moved the first paragraph of Section 1.3 to the end of Section 1.2. There are no references in the remained content of Section 1.3. 

We hope you are satisfied with this modification.

3. Paragraph one in Section 1.3 should be part of literature review and paragraph 2 of section should only be part of objectives section.

Thank you for this comment.

We have moved the first paragraph of Section 1.3 to the end of Section 1.2 according to your suggestion.

We hope you are satisfied with this revision.

---

## [Decision Letter · Decision Letter 5]

20 Dec 2024

A linear tessellation model for the identification of “food desert”: A case study of Shanghai, China

PONE-D-23-15204R5

Dear Dr. He,

We’re pleased to inform you that your manuscript has been judged scientifically suitable for publication and will be formally accepted for publication once it meets all outstanding technical requirements.

Kind regards,

Mohamed R. Abonazel, Ph.D.

Academic Editor

PLOS ONE

Additional Editor Comments (optional):

Authors should consider the comments of the third reviewer if possible but are not obliged to implement them.

Reviewers' comments:

Reviewer's Responses to Questions

**Comments to the Author**

1. If the authors have adequately addressed your comments raised in a previous round of review and you feel that this manuscript is now acceptable for publication, you may indicate that here to bypass the “Comments to the Author” section, enter your conflict of interest statement in the “Confidential to Editor” section, and submit your "Accept" recommendation.

Reviewer #3: (No Response)

2. Is the manuscript technically sound, and do the data support the conclusions?

Reviewer #3: Yes

3. Has the statistical analysis been performed appropriately and rigorously? 

Reviewer #3: Yes

4. Have the authors made all data underlying the findings in their manuscript fully available?

Reviewer #3: Yes

5. Is the manuscript presented in an intelligible fashion and written in standard English?

Reviewer #3: Yes

6. Review Comments to the Author

Reviewer #3: 1. In Section 3.4 Implication for Urban Planning, In this section proposed implications should be supported by arguments and data from the analysis not by reference to the published work as have been done in Lines 665, 671, 673, 676 have in text references.

7. PLOS authors have the option to publish the peer review history of their article (what does this mean?). If published, this will include your full peer review and any attached files.

Reviewer #3: **Yes: **Tejwant Singh Brar

---

## [Editor Report · Acceptance letter]

10 Jan 2025

PONE-D-23-15204R5 

PLOS ONE

Dear Dr. He, 

I'm pleased to inform you that your manuscript has been deemed suitable for publication in PLOS ONE. Congratulations! Your manuscript is now being handed over to our production team.

Kind regards, 

on behalf of

Dr Mohamed R. Abonazel 

Academic Editor

PLOS ONE